# INTENSITY-FREE LEARNING OF TEMPORAL POINT PROCESSES

**Oleksandr Shchur**,* **Marin Biloš**,* **Stephan Günnemann**
Technical University of Munich, Germany
{shchur,bilos,guennemann}@in.tum.de

## ABSTRACT

Temporal point processes are the dominant paradigm for modeling sequences of events happening at irregular intervals. The standard way of learning in such models is by estimating the conditional intensity function. However, parameterizing the intensity function usually incurs several trade-offs. We show how to overcome the limitations of intensity-based approaches by directly modeling the conditional distribution of inter-event times. We draw on the literature on normalizing flows to design models that are flexible and efficient. We additionally propose a simple mixture model that matches the flexibility of flow-based models, but also permits sampling and computing moments in closed form. The proposed models achieve state-of-the-art performance in standard prediction tasks and are suitable for novel applications, such as learning sequence embeddings and imputing missing data.

## 1 INTRODUCTION

Visits to hospitals, purchases in e-commerce systems, financial transactions, posts in social media — various forms of human activity can be represented as discrete events happening at irregular intervals. The framework of temporal point processes is a natural choice for modeling such data. By combining temporal point process models with deep learning, we can design algorithms able to learn complex behavior from real-world data.

Designing such models, however, usually involves trade-offs along the following dimensions: *flexibility* (can the model approximate any distribution?), *efficiency* (can the likelihood function be evaluated in closed form?), and *ease of use* (is sampling and computing summary statistics easy?). Existing methods (Du et al., 2016; Mei & Eisner, 2017; Omi et al., 2019) that are defined in terms of the conditional intensity function typically fall short in at least one of these categories.

Instead of modeling the intensity function, we suggest treating the problem of learning in temporal point processes as an instance of conditional density estimation. By using tools from neural density estimation (Bishop, 1994; Rezende & Mohamed, 2015), we can develop methods that have all of the above properties. To summarize, our contributions are the following:

- We connect the fields of temporal point processes and neural density estimation. We show how normalizing flows can be used to define flexible and theoretically sound models for learning in temporal point processes.
- We propose a simple mixture model that performs on par with the state-of-the-art methods. Thanks to its simplicity, the model permits closed-form sampling and moment computation.
- We show through a wide range of experiments how the proposed models can be used for prediction, conditional generation, sequence embedding and training with missing data.

## 2 BACKGROUND

**Definition.** A temporal point process (TPP) is a random process whose realizations consist of a sequence of strictly increasing arrival times $\mathcal{T} = \{t_1, ..., t_N\}$. A TPP can equivalently be represented

---

*Equal contribution
Code and datasets are available under https://github.com/shchur/ifl-tpp.

|  | Exponential intensity | Neural Hawkes | Fully NN | Normalizing Flows | Mixture Distribution |
|---|:---:|:---:|:---:|:---:|:---:|
| Closed-form likelihood | ✓ | ✗ | ✓ | ✓ | ✓ |
| Flexible | ✗ | ✓ | ✓ | ✓ | ✓ |
| Closed-form $\mathbb{E}[\tau]$ | ✗ | ✗ | ✗ | ✗ | ✓ |
| Closed-form sampling | ✓ | ✗ | ✗ | ✗ | ✓ |

Table 1: Comparison of neural temporal point process models that encode history with an RNN.

as a sequence of strictly positive inter-event times $\tau_i = t_i - t_{i-1} \in \mathbb{R}_+$. Representations in terms of $t_i$ and $\tau_i$ are isomorphic — we will use them interchangeably throughout the paper. The traditional way of specifying the dependency of the next arrival time $t$ on the history $\mathcal{H}_t = \{t_j \in \mathcal{T} : t_j < t\}$ is using the conditional intensity function $\lambda^*(t) := \lambda(t|\mathcal{H}_t)$. Here, the $*$ symbol reminds us of dependence on $\mathcal{H}_t$. Given the conditional intensity function, we can obtain the conditional probability density function (PDF) of the time $\tau_i$ until the next event by integration (Rasmussen, 2011) as $p^*(\tau_i) := p(\tau_i|\mathcal{H}_{t_i}) = \lambda^*(t_{i-1} + \tau_i) \exp\left(-\int_0^{\tau_i} \lambda^*(t_{i-1} + s)ds\right)$.

**Learning temporal point processes.** Conditional intensity functions provide a convenient way to specify point processes with a simple predefined behavior, such as self-exciting (Hawkes, 1971) and self-correcting (Isham & Westcott, 1979) processes. Intensity parametrization is also commonly used when learning a model from the data: Given a parametric intensity function $\lambda_{\boldsymbol{\theta}}^*(t)$ and a sequence of observations $\mathcal{T}$, the parameters $\boldsymbol{\theta}$ can be estimated by maximizing the log-likelihood: $\boldsymbol{\theta}^* = \arg\max_{\boldsymbol{\theta}} \sum_i \log p_{\boldsymbol{\theta}}^*(\tau_i) = \arg\max_{\boldsymbol{\theta}} \left[\sum_i \log \lambda_{\boldsymbol{\theta}}^*(t_i) - \int_0^{t_N} \lambda_{\boldsymbol{\theta}}^*(s)ds\right]$.

The main challenge of such intensity-based approaches lies in choosing a good parametric form for $\lambda_{\boldsymbol{\theta}}^*(t)$. This usually involves the following trade-off: For a "simple" intensity function (Du et al., 2016; Huang et al., 2019), the integral $\Lambda^*(\tau_i) := \int_0^{\tau_i} \lambda^*(t_{i-1}+s)ds$ has a closed form, which makes the log-likelihood easy to compute. However, such models usually have limited expressiveness. A more sophisticated intensity function (Mei & Eisner, 2017) can better capture the dynamics of the system, but computing log-likelihood will require approximating the integral using Monte Carlo.

Recently, Omi et al. (2019) proposed fully neural network intensity function (FullyNN) — a flexible, yet computationally tractable model for TPPs. The key idea of their approach is to model the cumulative conditional intensity function $\Lambda^*(\tau_i)$ using a neural network, which allows to efficiently compute the log-likelihood. Still, in its current state, the model has downsides: it doesn't define a valid PDF, sampling is expensive, and the expectation cannot be computed in closed form[1].

**This work.** We show that the drawbacks of the existing approaches can be remedied by looking at the problem of learning in TPPs from a different angle. Instead of modeling the conditional intensity $\lambda^*(t)$, we suggest to directly learn the conditional distribution $p^*(\tau)$. Modeling distributions with neural networks is a well-researched topic, that, surprisingly, is not usually discussed in the context of TPPs. By adopting this alternative point of view, we are able to develop new theoretically sound and effective methods (Section 3), as well as better understand the existing approaches (Section 4).

## 3 MODELS

We develop several approaches for modeling the distribution of inter-event times. First, we assume for simplicity that each inter-event time $\tau_i$ is conditionally independent of the history, given the model parameters (that is, $p^*(\tau_i) = p(\tau_i)$). In Section 3.1, we show how state-of-the-art neural density estimation methods based on normalizing flows can be used to model $p(\tau_i)$. Then in Section 3.2, we propose a simple mixture model that can match the performance of the more sophisticated flow-based models, while also addressing some of their shortcomings. Finally, we discuss how to make $p(\tau_i)$ depend on the history $\mathcal{H}_{t_i}$ in Section 3.3.

---

[1] A more detailed discussion of the FullyNN model follows in Section 4 and Appendix C.

## 3.1 Modeling $p(\tau)$ with normalizing flows

The core idea of normalizing flows (Tabak & Turner, 2013; Rezende & Mohamed, 2015) is to define a flexible probability distribution by transforming a simple one. Assume that $z$ has a PDF $q(z)$. Let $x = g(z)$ for some differentiable invertible transformation $g : \mathcal{Z} \to \mathcal{X}$ (where $\mathcal{Z}, \mathcal{X} \subseteq \mathbb{R})^2$. We can obtain the PDF $p(x)$ of $x$ using the change of variables formula as $p(x) = q(g^{-1}(x)) \left| \frac{\partial}{\partial x} g^{-1}(x) \right|$. By stacking multiple transformations $g_1, ..., g_M$, we obtain an expressive probability distribution $p(x)$. To draw a sample $x \sim p(x)$, we need to draw $z \sim q(z)$ and compute the *forward* transformation $x = (g_M \circ \cdots \circ g_1)(z)$. To get the density of an arbitrary point $x$, it is necessary to evaluate the *inverse* transformation $z = (g_1^{-1} \circ \cdots \circ g_M^{-1})(x)$ and compute $q(z)$. Modern normalizing flows architectures parametrize the transformations using extremely flexible functions $f_{\boldsymbol{\theta}}$, such as polynomials (Jaini et al., 2019) or neural networks (Krueger et al., 2018). The flexibility of these functions comes at a cost — while the inverse $f_{\boldsymbol{\theta}}^{-1}$ exists, it typically doesn't have a closed form. That is, if we use such a function to define one direction of the transformation in a flow model, the other direction can only be approximated numerically using iterative root-finding methods (Ho et al., 2019). In this work, we don't consider invertible normalizing flows based on dimension splitting, such as RealNVP (Dinh et al., 2017), since they are not applicable to 1D data.

In the context of TPPs, our goal is to model the distribution $p(\tau)$ of inter-event times. In order to be able to learn the parameters of $p(\tau)$ using maximum likelihood, we need to be able to evaluate the density at any point $\tau$. For this we need to define the inverse transformation $g^{-1} := (g_1^{-1} \circ \cdots \circ g_M^{-1})$. First, we set $z_M = g_M^{-1}(\tau) = \log \tau$ to convert a positive $\tau \in \mathbb{R}_+$ into $z_M \in \mathbb{R}$. Then, we stack multiple layers of parametric functions $f_{\boldsymbol{\theta}} : \mathbb{R} \to \mathbb{R}$ that can approximate any transformation. We consider two choices for $f_{\boldsymbol{\theta}}$: deep sigmoidal flow (DSF) from Krueger et al. (2018) and sum-of-squares (SOS) polynomial flow from Jaini et al. (2019)

$$f^{DSF}(x) = \sigma^{-1} \left( \sum_{k=1}^{K} w_k \sigma \left( \frac{x - \mu_k}{s_k} \right) \right) \quad f^{SOS}(x) = a_0 + \sum_{k=1}^{K} \sum_{p=0}^{R} \sum_{q=0}^{R} \frac{a_{p,k} a_{q,k}}{p + q + 1} x^{p+q+1} \quad (1)$$

where $\boldsymbol{a}, \boldsymbol{w}, \boldsymbol{s}, \boldsymbol{\mu}$ are the transformation parameters, $K$ is the number of components, $R$ is the polynomial degree, and $\sigma(x) = 1/(1 + e^{-x})$. We denote the two variants of the model based on $f^{DSF}$ and $f^{SOS}$ building blocks as **DSFlow** and **SOSFlow** respectively. Finally, after stacking multiple $g_m^{-1} = f_{\boldsymbol{\theta}_m}$, we apply a sigmoid transformation $g_1^{-1} = \sigma$ to convert $z_2$ into $z_1 \in (0, 1)$.

For both models, we can evaluate the inverse transformations $(g_1^{-1} \circ \cdots \circ g_M^{-1})$, which means the model can be efficiently trained via maximum likelihood. The density $p(\tau)$ defined by either DSFlow or SOSFlow model is extremely flexible and can approximate any distribution (Section 3.4). However, for some use cases, this is not sufficient. For example, we may be interested in the expected time until the next event, $\mathbb{E}_p[\tau]$. In this case, flow-based models are not optimal, since for them $\mathbb{E}_p[\tau]$ does not in general have a closed form. Moreover, the forward transformation $(g_M \circ \cdots \circ g_1)$ cannot be computed in closed form since the functions $f^{DSF}$ and $f^{SOS}$ cannot be inverted analytically. Therefore, sampling from $p(\tau)$ is also problematic and requires iterative root finding.

This raises the question: Can we design a model for $p(\tau)$ that is as expressive as the flow-based models, but in which sampling and computing moments is easy and can be done in closed form?

## 3.2 Modeling $p(\tau)$ with mixture distributions

**Model definition.** While mixture models are commonly used for clustering, they can also be used for density estimation. Mixtures work especially well in low dimensions (McLachlan & Peel, 2004), which is the case in TPPs, where we model the distribution of one-dimensional inter-event times $\tau$. Since the inter-event times $\tau$ are positive, we choose to use a mixture of log-normal distributions to model $p(\tau)$. The PDF of a log-normal mixture is defined as

$$p(\tau | \boldsymbol{w}, \boldsymbol{\mu}, \boldsymbol{s}) = \sum_{k=1}^{K} w_k \frac{1}{\tau s_k \sqrt{2\pi}} \exp \left( -\frac{(\log \tau - \mu_k)^2}{2 s_k^2} \right) \quad (2)$$

---

[2] All definitions can be extended to $\mathbb{R}^D$ for $D > 1$. We consider the one-dimensional case since our goal is to model the distribution of inter-event times $\tau \in \mathbb{R}_+$.

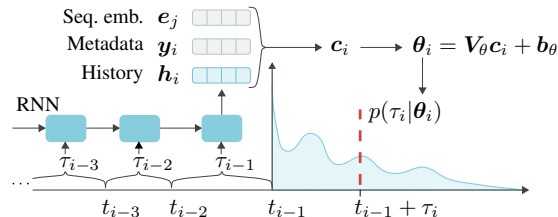
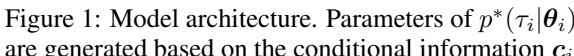

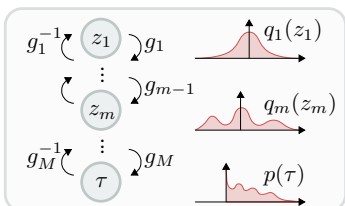

Figure 1: Model architecture. Parameters of $p^*(\tau_i|\boldsymbol{\theta}_i)$ are generated based on the conditional information $\boldsymbol{c}_i$.

Figure 2: Normalizing flows define a flexible distribution via transformations.

where $\boldsymbol{w}$ are the mixture weights, $\boldsymbol{\mu}$ are the mixture means, and $\boldsymbol{s}$ are the standard deviations. Because of its simplicity, the log-normal mixture model has a number of attractive properties.

**Moments.** Since each component $k$ has a finite mean, the mean of the entire distribution can be computed as $\mathbb{E}_p[\tau] = \sum_k w_k \exp(\mu_k + s_k^2/2)$, i.e., a weighted average of component means. Higher moments can be computed based on the moments of each component (Frühwirth-Schnatter, 2006).

**Sampling.** While flow-based models from Section 3.1 require iterative root-finding algorithms to generate samples, sampling from a mixture model can be done in closed form:

$$\boldsymbol{z} \sim \text{Categorical}(\boldsymbol{w}) \qquad \varepsilon \sim \text{Normal}(0,1) \qquad \tau = \exp(\boldsymbol{s}^T\boldsymbol{z} \cdot \varepsilon + \boldsymbol{\mu}^T\boldsymbol{z})$$

where $\boldsymbol{z}$ is a one-hot vector of size $K$. In some applications, such as reinforcement learning (Upadhyay et al., 2018), we might be interested in computing gradients of the samples w.r.t. the model parameters. The samples $\tau$ drawn using the procedure above are differentiable with respect to the means $\boldsymbol{\mu}$ and scales $\boldsymbol{s}$. By using the Gumbel-softmax trick (Jang et al., 2017) when sampling $\boldsymbol{z}$, we can obtain gradients w.r.t. all the model parameters (Appendix D.6). Such reparametrization gradients have lower variance and are easier to implement than the score function estimators typically used in other works (Mohamed et al., 2019). Other flexible models (such as multi-layer flow models from Section 3.1) do not permit sampling through reparametrization, and thus are not well-suited for the above-mentioned scenario. In Section 5.4, we show how reparametrization sampling can also be used to train with missing data by performing imputation on the fly.

## 3.3 INCORPORATING THE CONDITIONAL INFORMATION

**History.** A crucial feature of temporal point processes is that the time $\tau_i = (t_i - t_{i-1})$ until the next event may be influenced by all the events that happened before. A standard way of capturing this dependency is to process the event history $\mathcal{H}_{t_i}$ with a recurrent neural network (RNN) and embed it into a fixed-dimensional vector $\boldsymbol{h}_i \in \mathbb{R}^H$ (Du et al., 2016).

**Conditioning on additional features.** The distribution of the time until the next event might depend on factors other than the history. For instance, distribution of arrival times of customers in a restaurant depends on the day of the week. As another example, if we are modeling user behavior in an online system, we can obtain a different distribution $p^*(\tau)$ for each user by conditioning on their metadata. We denote such side information as a vector $\boldsymbol{y}_i$. Such information is different from marks (Rasmussen, 2011), since (a) the metadata may be shared for the entire sequence and (b) $\boldsymbol{y}_i$ only influences the distribution $p^*(\tau_i|\boldsymbol{y}_i)$, not the objective function.

In some scenarios, we might be interested in learning from multiple event sequences. In such case, we can assign each sequence $\mathcal{T}_j$ a learnable **sequence embedding** vector $\boldsymbol{e}_j$. By optimizing $\boldsymbol{e}_j$, the model can learn to distinguish between sequences that come from different distributions. The learned embeddings can then be used for visualization, clustering or other downstream tasks.

**Obtaining the parameters.** We model the conditional dependence of the distribution $p^*(\tau_i)$ on all of the above factors in the following way. The history embedding $\boldsymbol{h}_i$, metadata $\boldsymbol{y}_i$ and sequence embedding $\boldsymbol{e}_j$ are concatenated into a context vector $\boldsymbol{c}_i = [\boldsymbol{h}_i||\boldsymbol{y}_i||\boldsymbol{e}_j]$. Then, we obtain the parameters of the distribution $p^*(\tau_i)$ as an affine function of $\boldsymbol{c}_i$. For example, for the mixture model we have

$$\boldsymbol{w}_i = \text{softmax}(\boldsymbol{V_w}\boldsymbol{c}_i + \boldsymbol{b_w}) \qquad \boldsymbol{s}_i = \exp(\boldsymbol{V_s}\boldsymbol{c}_i + \boldsymbol{b_s}) \qquad \boldsymbol{\mu}_i = \boldsymbol{V_\mu}\boldsymbol{c}_i + \boldsymbol{b_\mu} \qquad (3)$$

where the softmax and exp transformations are applied to enforce the constraints on the distribution parameters, and $\{\boldsymbol{V_w}, \boldsymbol{V_s}, \boldsymbol{V_\mu}, \boldsymbol{b_w}, \boldsymbol{b_s}, \boldsymbol{b_\mu}\}$ are learnable parameters. Such model resembles the

mixture density network architecture (Bishop, 1994). The whole process is illustrated in Figure 1. We obtain the parameters of the flow-based models in a similar way (see Appendix D).

## 3.4 DISCUSSION

**Universal approximation.** The SOSFlow and DSFlow models can approximate any probability density on $\mathbb{R}$ arbitrarily well (Jaini et al., 2019, Theorem 3), (Krueger et al., 2018, Theorem 4). It turns out, a mixture model has the same universal approximation (UA) property.

**Theorem 1** *(DasGupta, 2008, Theorem 33.2). Let $p(x)$ be a continuous density on $\mathbb{R}$. If $q(x)$ is any density on $\mathbb{R}$ and is also continuous, then, given $\varepsilon > 0$ and a compact set $\mathcal{S} \subset \mathbb{R}$, there exist number of components $K \in \mathbb{N}$, mixture coefficients $\boldsymbol{w} \in \Delta^{K-1}$, locations $\boldsymbol{\mu} \in \mathbb{R}^K$, and scales $\boldsymbol{s} \in \mathbb{R}_+^K$ such that for the mixture distribution $\hat{p}(x) = \sum_{k=1}^{K} w_k \frac{1}{s_k} q(\frac{x-\mu_k}{s_k})$ it holds $\sup_{x \in \mathcal{S}} |p(x) - \hat{p}(x)| < \varepsilon$.*

This results shows that, in principle, the mixture distribution is as expressive as the flow-based models. Since we are modeling the conditional density, we additionally need to assume for all of the above models that the RNN can encode all the relevant information into the history embedding $\boldsymbol{h}_i$. This can be accomplished by invoking the universal approximation theorems for RNNs (Siegelmann & Sontag, 1992; Schäfer & Zimmermann, 2006).

Note that this result, like other UA theorems of this kind (Cybenko, 1989; Daniels & Velikova, 2010), does not provide any practical guarantees on the obtained approximation quality, and doesn't say how to learn the model parameters. Still, UA intuitively seems like a desirable property of a distribution. This intuition is supported by experimental results. In Section 5.1, we show that models with the UA property consistently outperform the less flexible ones.

Interestingly, Theorem 1 does not make any assumptions about the form of the base density $q(x)$. This means we could as well use a mixture of distribution other than log-normal. However, other popular distributions on $\mathbb{R}_+$ have drawbacks: log-logistic does not always have defined moments and gamma distribution doesn't permit straightforward sampling with reparametrization.

**Intensity function.** For both flow-based and mixture models, the conditional cumulative distribution function (CDF) $F^*(\tau)$ and the PDF $p^*(\tau)$ are readily available. This means we can easily compute the respective intensity functions (see Appendix A). However, we should still ask whether we lose anything by modeling $p^*(\tau)$ instead of $\lambda^*(t)$. The main arguments in favor of modeling the intensity function in traditional models (e.g. self-exciting process) are that it's *intuitive*, *easy to specify* and *reusable* (Upadhyay & Rodriguez, 2019).

"Intensity function is *intuitive*, while the conditional density is not." — While it's true that in simple models (e.g. in self-exciting or self-correcting processes) the dependence of $\lambda^*(t)$ on the history is intuitive and interpretable, modern RNN-based intensity functions (as in Du et al. (2016); Mei & Eisner (2017); Omi et al. (2019)) cannot be easily understood by humans. In this sense, our proposed models are as intuitive and interpretable as other existing intensity-based neural network models.

"$\lambda^*(t)$ is *easy to specify*, since it only has to be positive. On the other hand, $p^*(\tau)$ must integrate to one." — As we saw, by using either normalizing flows or a mixture distribution, we automatically enforce that the PDF integrates to one, without sacrificing the flexibility of our model.

"*Reusability:* If we merge two independent point processes with intensitites $\lambda_1^*(t)$ and $\lambda_2^*(t)$, the merged process has intensity $\lambda^*(t) = \lambda_1^*(t) + \lambda_2^*(t)$." — An equivalent result exists for the CDFs $F_1^*(\tau)$ and $F_2^*(\tau)$ of the two independent processes. The CDF of the merged process is obtained as $F^*(\tau) = F_1^*(\tau) + F_2^*(\tau) - F_1^*(\tau)F_2^*(\tau)$ (derivation in Appendix A).

As we just showed, modeling $p^*(\tau)$ instead of $\lambda^*(t)$ does not impose any limitation on our approach. Moreover, a mixture distribution is flexible, easy to sample from and has well-defined moments, which favorably compares it to other intensity-based deep learning models.

## 4 RELATED WORK

**Neural temporal point processes.** Fitting simple TPP models (e.g. self-exciting (Hawkes, 1971) or self-correcting (Isham & Westcott, 1979) processes) to real-world data may lead to poor results because of model misspecification. Multiple recent works address this issue by proposing more

flexible neural-network-based point process models. These neural models are usually defined in terms of the conditional intensity function. For example, Mei & Eisner (2017) propose a novel RNN architecture that can model sophisticated intensity functions. This flexibility comes at the cost of inability to evaluate the likelihood in closed form, and thus requiring Monte Carlo integration.

Du et al. (2016) suggest using an RNN to encode the event history into a vector $\boldsymbol{h}_i$. The history embedding $\boldsymbol{h}_i$ is then used to define the conditional intensity, for example, using the *constant intensity* model $\lambda^*(t_i) = \exp(\boldsymbol{v}^T \boldsymbol{h}_i + b)$ (Li et al., 2018; Huang et al., 2019) or the more flexible *exponential intensity* model $\lambda^*(t_i) = \exp(w(t_i - t_{i-1}) + \boldsymbol{v}^T \boldsymbol{h}_i + b)$ (Du et al., 2016; Upadhyay et al., 2018). By considering the conditional distribution $p^*(\tau)$ of the two models, we can better understand their properties. Constant intensity corresponds to an exponential distribution, and exponential intensity corresponds to a Gompertz distribution (see Appendix B). Clearly, these unimodal distributions cannot match the flexibility of a mixture model (as can be seen in Figure 8).

Omi et al. (2019) introduce a flexible fully neural network (FullyNN) intensity model, where they model the cumulative intensity function $\Lambda^*(\tau)$ with a neural net. The function $\Lambda^*$ converts $\tau$ into an exponentially distributed random variable with unit rate (Rasmussen, 2011), similarly to how normalizing flows model $p^*(\tau)$ by converting $\tau$ into a random variable with a simple distribution. However, due to a suboptimal choice of the network architecture, the PDF of the FullyNN model does not integrate to 1, and the model assigns non-zero probability to *negative* inter-event times (see Appendix C). In contrast, SOSFlow and DSFlow always define a valid PDF on $\mathbb{R}_+$. Moreover, similar to other flow-based models, sampling from the FullyNN model requires iterative root finding.

Several works used mixtures of kernels to parametrize the conditional intensity function (Taddy et al., 2012; Tabibian et al., 2017; Okawa et al., 2019). Such models can only capture self-exciting influence from past events. Moreover, these models do not permit computing expectation and drawing samples in closed form. Recently, Biloš et al. (2019) and Türkmen et al. (2019) proposed neural models for learning marked TPPs. These models focus on event type prediction and share the limitations of other neural intensity-based approaches. Other recent works consider alternatives to the maximum likelihood objective for training TPPs. Examples include noise-contrastive estimation (Guo et al., 2018), Wasserstein distance (Xiao et al., 2017; 2018; Yan et al., 2018), and reinforcement learning (Li et al., 2018; Upadhyay et al., 2018). This line of research is orthogonal to our contribution, and the models proposed in our work can be combined with the above-mentioned training procedures.

**Neural density estimation.** There exist two popular paradigms for learning flexible probability distributions using neural networks: In mixture density networks (Bishop, 1994), a neural net directly produces the distribution parameters; in normalizing flows (Tabak & Turner, 2013; Rezende & Mohamed, 2015), we obtain a complex distribution by transforming a simple one. Both mixture models (Schuster, 2000; Eirola & Lendasse, 2013; Graves, 2013) and normalizing flows (Oord et al., 2016; Ziegler & Rush, 2019) have been applied for modeling sequential data. However, surprisingly, none of the existing works make the connection and consider these approaches in the context of TPPs.

## 5 EXPERIMENTS

We evaluate the proposed models on the established task of event time prediction (with and without marks) in Sections 5.1 and 5.2. In the remaining experiments, we show how the log-normal mixture model can be used for incorporating extra conditional information, training with missing data and learning sequence embeddings. We use 6 **real-world datasets** containing event data from various domains: Wikipedia (article edits), MOOC (user interaction with online course system), Reddit (posts in social media) (Kumar et al., 2019), Stack Overflow (badges received by users), LastFM (music playback) (Du et al., 2016), and Yelp (check-ins to restaurants). We also generate 5 **synthetic datasets** (Poisson, Renewal, Self-correcting, Hawkes1, Hawkes2), as described in Omi et al. (2019). Detailed descriptions and summary statistics of all the datasets are provided in Appendix E.

### 5.1 EVENT TIME PREDICTION USING HISTORY

**Setup.** We consider two normalizing flow models, **SOSFlow** and **DSFlow** (Equation 1), as well a log-normal mixture model (Equation 2), denoted as **LogNormMix**. As baselines, we consider **RMTPP** (i.e. Gompertz distribution / exponential intensity from Du et al. (2016)) and **FullyNN**

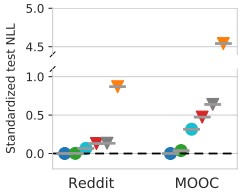

Figure 3: NLL loss for event time prediction without marks (left) and with marks (right). NLL of each model is standardized by subtracting the score of LogNormMix. **Lower score is better.** Despite its simplicity, LogNormMix consistently achieves excellent loss values.

model by Omi et al. (2019). Additionally, we use a single log-normal distribution (denoted **LogNormal**) to highlight the benefits of the mixture model. For all models, an RNN encodes the history into a vector $h_i$. The parameters of $p^*(\tau)$ are then obtained using $h_i$ (Equation 3). We exclude the NeuralHawkes model from our comparison, since it is known to be inferior to RMTPP in time prediction (Mei & Eisner, 2017), and, unlike other models, doesn't have a closed-form likelihood.

Each dataset consists of multiple sequences of event times. The task is to predict the time $\tau_i$ until the next event given the history $\mathcal{H}_{t_i}$. For each dataset, we use 60% of the sequences for training, 20% for validation and 20% for testing. We train all models by minimizing the negative log-likelihood (NLL) of the inter-event times in the training set. To ensure a fair comparison, we try multiple hyperparameter configurations for each model and select the best configuration using the validation set. Finally, we report the NLL loss of each model on the test set. All results are averaged over 10 train/validation/test splits. Details about the implementation, training process and hyperparameter ranges are provided in Appendix D. For each real-world dataset, we report the difference between the NLL loss of each method and the LogNormMix model (Figure 3). We report the *differences*, since scores of all models can be shifted arbitrarily by scaling the data. Absolute scores (not differences) in a tabular format, as well as results for synthetic datasets are provided in Appendix F.1.

**Results.** Simple unimodal distributions (Gompertz/RMTPP, LogNormal) are always dominated by the more flexible models with the universal approximation property (LogNormMix, DSFlow, SOS-Flow, FullyNN). Among the simple models, LogNormal provides a much better fit to the data than RMTPP/Gompertz. The distribution of inter-event times in real-world data often has heavy tails, and the Gompertz distributions fails to capture this behavior. We observe that the two proposed models, LogNormMix and DSFlow consistently achieve the best loss values.

## 5.2 Learning with marks

**Setup.** We apply the models for learning in *marked* temporal point processes. Marks are known to improve performance of simpler models (Du et al., 2016), we want to establish whether our proposed models work well in this setting. We use the same setup as in the previous section, except for two differences. The RNN takes a tuple $(\tau_i, m_i)$ as input at each time step, where $m_i$ is the mark. Moreover, the loss function now includes a term for predicting the next mark: $\mathcal{L}(\boldsymbol{\theta}) = -\sum_i \left[\log p_{\boldsymbol{\theta}}^*(\tau_i) + \log p_{\boldsymbol{\theta}}^*(m_i)\right]$ (implementation details in Appendix F.2).

**Results.** Figure 3 (right) shows the time NLL loss (i.e. $-\sum_i \log p^*(\tau_i)$) for Reddit and MOOC datasets. LogNormMix shows dominant performance in the marked case, just like in the previous experiment. Like before, we provide the results in tabular format, as well as report the marks NLL loss in Appendix F.

## 5.3 Learning with additional conditional information

**Setup.** We investigate whether the additional conditional information (Section 3.3) can improve performance of the model. In the Yelp dataset, the task is predict the time $\tau$ until the next check-in for a given restaurant. We postulate that the distribution $p^*(\tau)$ is different, depending on whether it's a weekday and whether it's an evening hour, and encode this information as a vector $y_i$. We consider 4 variants of the LogNormMix model, that either use or don't use $y_i$ and the history embedding $h_i$.

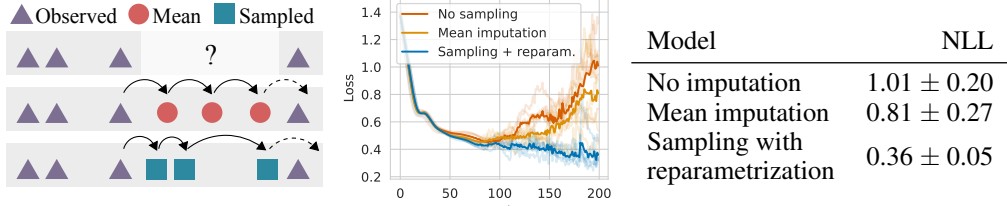

Figure 4: By sampling the missing values from $p^*(\tau)$ during training, LogNormMix learns the true underlying data distribution. Other imputation strategies overfit the partially observed sequence.

**Results.** Figure 5 shows the test set loss for 4 variants of the model. We see that additional conditional information boosts performance of the LogNormMix model, regardless of whether the history embedding is used.

### 5.4    MISSING DATA IMPUTATION

In practical scenarios, one often has to deal with missing data. For example, we may know that records were not kept for a period of time, or that the data is unusable for some reason. Since TPPs are a generative model, they provide a principled way to handle the missing data through imputation.

**Setup.** We are given several sequences generated by a Hawkes process, where some parts are known to be missing. We consider 3 strategies for learning from such a partially observed sequence: (a) ignore the gaps, maximize log-likelihood of observed inter-event times (b) fill the gaps with the average $\tau$ estimated from observed data, maximize log-likelihood of observed data, and (c) fill the gaps with samples generated by the model, maximize the *expected* log-likelihood of the observed points. The setup is demonstrated in Figure 4. Note that in case (c) the expected value depends on the parameters of the distribution, hence we need to perform sampling with reparametrization to optimize such loss. A more detailed description of the setup is given in Appendix F.4.

**Results.** The 3 model variants are trained on the partially-observed sequence. Figure 4 shows the NLL of the fully observed sequence (not seen by any model at training time) produced by each strategy. We see that strategies (a) and (b) overfit the partially observed sequence. In contrast, strategy (c) generalizes and learns the true underlying distribution. The ability of the LogNormMix model to draw samples with reparametrization was crucial to enable such training procedure.

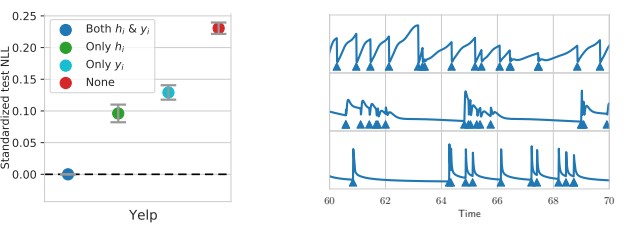

Figure 5: Conditional information improves performance.

Figure 6: Sequences generated based on different embeddings.

Figure 7: Sequence embeddings learned by the model.

### 5.5    SEQUENCE EMBEDDING

Different sequences in the dataset might be generated by different processes, and exhibit different distribution of inter-event times. We can "help" the model distinguish between them by assigning a trainable embedding vector $e_j$ to each sequence $j$ in the dataset. It seems intuitive that embedding vectors learned this way should capture some notion of similarity between sequences.

**Learned sequence embeddings.** We learn a sequence embedding for each of the sequences in the synthetic datasets (along with other model parameters). We visualize the learned embeddings using t-SNE (Maaten & Hinton, 2008) in Figure 7 colored by the true class. As we see, the model learns to differentiate between sequences from different distributions in a completely unsupervised way.

**Generation.** We fit the LogNormMix model to two sequences (from self-correcting and renewal processes), and, respectively, learn two embedding vectors $e_{SC}$ and $e_{RN}$. After training, we generate 3 sequences from the model, using $e_{SC}$, $1/2(e_{SC} + e_{RN})$ and $e_{RN}$ as sequence embeddings. Additionally, we plot the *learned* conditional intensity function of our model for each generated sequence (Figure 6). The model learns to map the sequence embeddings to very different distributions.

## 6 CONCLUSIONS

We use tools from neural density estimation to design new models for learning in TPPs. We show that a simple mixture model is competitive with state-of-the-art normalizing flows methods, as well as convincingly outperforms other existing approaches. By looking at learning in TPPs from a different perspective, we were able to address the shortcomings of existing intensity-based approaches, such as insufficient flexibility, lack of closed-form likelihoods and inability to generate samples analytically. We hope this alternative viewpoint will inspire new developments in the field of TPPs.

## ACKNOWLEDGMENTS

This research was supported by the German Federal Ministry of Education and Research (BMBF), grant no. 01IS18036B, and the Software Campus Project Deep-RENT. The authors of this work take full responsibilities for its content.

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

## A   INTENSITY FUNCTION OF FLOW AND MIXTURE MODELS

**CDF and conditional intensity function of proposed models.** The cumulative distribution function (CDF) of a **normalizing flow** model can be obtained in the following way. If $z$ has a CDF $Q(z)$ and $\tau = g(z)$, then the CDF $F(\tau)$ of $\tau$ is obtained as

$$F(\tau) = Q(g^{-1}(\tau))$$

Since for both SOSFlow and DSFlow we can evaluate $g^{-1}$ in closed form, $F(\tau)$ is easy to compute.

For the **log-normal mixture model**, CDF is by definition equal to

$$F(\tau) = \sum_{k=1}^{K} w_k \Phi \left( \frac{\log \tau - \mu_k}{s_k} \right)$$

where $\Phi(\cdot)$ is the CDF of a standard normal distribution.

Given the conditional PDF and CDF, we can compute the conditional intensity $\lambda^*(t)$ and the cumulative intensity $\Lambda^*(\tau)$ for each model as

$$\lambda^*(t) = \frac{p^*(t - t_{i-1})}{1 - F^*(t - t_{i-1})} \qquad \Lambda^*(\tau_i) := \int_0^{\tau_i} \lambda^*(t_{i-1} + s)ds = -\log(1 - F^*(\tau_i))$$

where $t_{i-1}$ is the arrival time of most recent event before $t$ (Rasmussen, 2011).

**Merging two independent processes.** We replicate the setup from Upadhyay & Rodriguez (2019) and consider what happens if we merge two *independent* TPPs with intensity functions $\lambda_1^*(t)$ and $\lambda_2^*(t)$ (and respectively, cumulative intensity functions $\Lambda_1^*(\tau)$ and $\Lambda_2^*(\tau)$). According to Upadhyay & Rodriguez (2019), the intensity function of the new process is $\lambda^*(t) = \lambda_1^*(t) + \lambda_2^*(t)$. Therefore, the cumulative intensity function of the new process is

$$\begin{aligned}
\Lambda^*(\tau) &= \int_0^\tau \lambda^*(t_{i-1} + s)ds \\
&= \int_0^\tau \lambda_1^*(t_{i-1} + s)ds + \int_0^\tau \lambda_2^*(t_{i-1} + s)ds \\
&= \Lambda_1^*(\tau) + \Lambda_2^*(\tau)
\end{aligned}$$

Using the previous result, we can obtain the CDF of the merged process as

$$\begin{aligned}
F^*(\tau) &= 1 - \exp(-\Lambda^*(\tau)) \\
&= 1 - \exp(-\Lambda_1^*(\tau) - \Lambda_2^*(\tau)) \\
&= 1 - \exp(\log(1 - F_1^*(\tau)) + \log(1 - F_2^*(\tau))) \\
&= 1 - (1 + F_1^*(\tau)F_2^*(\tau) - F_1^*(\tau) - F_2^*(\tau)) \\
&= F_1^*(\tau) + F_2^*(\tau) - F_1^*(\tau)F_2^*(\tau)
\end{aligned}$$

The PDF of the merged process is obtained by simply differentiating the CDF w.r.t. $\tau$.

This means that by using either normalizing flows or mixture distributions, and thus directly modeling PDF / CDF, we are not losing any benefits of the intensity parametrization.

## B   DISCUSSION OF CONSTANT & EXPONENTIAL INTENSITY MODELS

**Constant intensity model as exponential distribution.** The conditional intensity function of the constant intensity model (Upadhyay et al., 2018) is defined as $\lambda^*(t_i) = \exp(\boldsymbol{v}^T \boldsymbol{h}_i + b)$, where $\boldsymbol{h}_i \in \mathbb{R}^H$ is the history embedding produced by an RNN, and $b \in \mathbb{R}$ is a learnable parameter. By setting $c = \exp(\boldsymbol{v}^T \boldsymbol{h}_i + b)$, it's easy to see that the PDF of the constant intensity model $p^*(\tau) = c \exp(-c)$ corresponds to an exponential distribution.

**Exponential intensity model as Gompertz distribution.**   PDF of a Gompertz distribution (Wienke, 2010) is defined as

$$p(\tau|\alpha, \beta) = \alpha \exp \left( \beta\tau - \frac{\alpha}{\beta} \exp(\beta t) + \frac{\alpha}{\beta} \right)$$

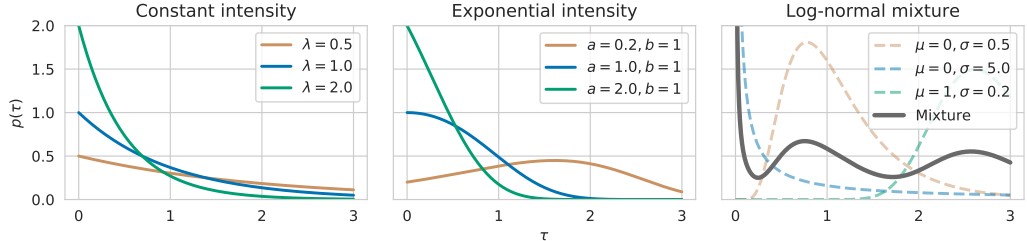

Figure 8: Different choices for modeling $p(\tau)$: exponential distribution (left), Gompertz distribution (center), log-normal mixture (right). Mixture distribution can approximate any density while being tractable and easy to sample from.

for $\alpha, \beta > 0$. The two parameters $\alpha$ and $\beta$ define its shape and rate, respectively. For any choice of its parameters, Gompertz distribution is unimodal and light-tailed. The mean of the Gompertz distribution can be computed as $\mathbb{E}[\tau] = \frac{1}{\beta} \exp\left(\frac{\alpha}{\beta}\right) \text{Ei}(-\frac{\alpha}{\beta})$, where $\text{Ei}(z) = \int_{-z}^{\infty} \exp(-v)/v \, dv$ is the exponential integral function (that can be approximated numerically).

The conditional intensity function of the exponential intensity model (Du et al., 2016) is defined as $\lambda^*(t_i) = \exp(w(t_i - t_{i-1}) + \boldsymbol{v}^T \boldsymbol{h}_i + b)$, where $\boldsymbol{h}_i \in \mathbb{R}^H$ is the history embedding produced by an RNN, and $\boldsymbol{v} \in \mathbb{R}^H, b \in \mathbb{R}, w \in \mathbb{R}_+$ are learnable parameters. By defining $d = \boldsymbol{v}^T \boldsymbol{h}_i + b$, we obtain the PDF of the exponential intensity model (Du et al., 2016, Equation 12) as

$$p(\tau|w, d) = \exp\left(w\tau + d - \frac{1}{w}\exp(w\tau + d) + \frac{1}{w}\exp(d)\right)$$

By setting $\alpha = \exp(d)$ and $\beta = w$ we see that the exponential intensity model is equivalent to a Gompertz distribution.

**Discussion.** Figure 8 shows densities that can be represented by exponential and Gompertz distributions. Even though the history embedding $\boldsymbol{h}_i$ produced by an RNN may capture rich information, the resulting distribution $p^*(\tau_i)$ for both models has very limited flexibility, is unimodal and light-tailed. In contrast, a flow-based or a mixture model is significantly more flexible and can approximate any density.

## C    DISCUSSION OF THE FULLYNN MODEL

**Summary**    The main idea of the approach by Omi et al. (2019) is to model the integrated conditional intensity function

$$\Lambda^*(\tau) = \int_0^\tau \lambda^*(t_{i-1} + s)ds$$

using a feedforward neural network with non-negative weights

$$\Lambda^*(\tau) := f(\tau) = \text{softplus}(\boldsymbol{W}^{(3)}\tanh(\boldsymbol{W}^{(2)}\tanh(\boldsymbol{W}^{(1)}\tau + \tilde{\boldsymbol{b}}^{(1)}) + \boldsymbol{b}^{(2)}) + \boldsymbol{b}^{(3)}) \qquad (4)$$

where $\tilde{\boldsymbol{b}}^{(1)} = \boldsymbol{V}\boldsymbol{h} + \boldsymbol{b}^{(0)}$, $\boldsymbol{h} \in \mathbb{R}^H$ is the history embedding, $\boldsymbol{W}^{(1)} \in \mathbb{R}_+^{D \times 1}$, $\boldsymbol{W}^{(2)} \in \mathbb{R}_+^{D \times D}$, $\boldsymbol{W}^{(3)} \in \mathbb{R}_+^{1 \times D}$ are non-negative weight matrices, and $\boldsymbol{V} \in \mathbb{R}^{D \times H}$, $\boldsymbol{b}^{(0)} \in \mathbb{R}^D$, $\boldsymbol{b}^{(2)} \in \mathbb{R}^D$, $\boldsymbol{b}^{(3)} \in \mathbb{R}$ are the remaining model parameters.

**FullyNN as a normalizing flow**    Let $z \sim \text{Exponential}(1)$, that is

$$F(z) = 1 - \exp(-z) \qquad\qquad p(z) = \exp(-z)$$

We can view $f : \mathbb{R}_+ \to \mathbb{R}_+$ as a transformation that maps $\tau$ to $z$

$$z = f(\tau) \iff \tau = f^{-1}(z)$$

We can now use the change of variables formula to obtain the conditional CDF and PDF of $\tau$.

Alternatively, we can obtain the conditional intensity as

$$\lambda^*(\tau) = \frac{\partial}{\partial \tau} \Lambda^*(\tau) = \frac{\partial}{\partial \tau} f(\tau)$$

and use the fact that $p^*(\tau_i) = \lambda^*(t_{i-1} + \tau_i) \exp\left(-\int_0^{\tau_i} \lambda^*(t_{i-1} + s)ds\right)$.

Both approaches lead to the same conclusion

$$F^*(\tau) = 1 - \exp(-f(\tau)) \qquad\qquad p^*(\tau) = \exp(-f(\tau))\frac{\partial}{\partial \tau} f(\tau)$$

However, the first approach also provides intuition on how to draw samples $\tilde{\tau}$ from the resulting distribution $p^*(\tau)$ — an approach known as the inverse method (Rasmussen, 2011)

1. Sample $\tilde{z} \sim \text{Exponential}(1)$
2. Obtain $\tilde{\tau}$ by solving $f(\tau) - \tilde{z} = 0$ for $\tau$ (using e.g. bisection method)

Similarly to other flow-based models, sampling from the FullyNN model cannot be done exactly and requires a numerical approximation.

**Shortcomings of the FullyNN model**

1. The PDF defined by the FullyNN model doesn't integrate to 1.

   By definition of the CDF, the condition that the PDF integrates to 1 is equivalent to $\lim_{\tau \to \infty} F^*(\tau) = 1$, which in turn is equivalent to $\lim_{\tau \to \infty} \Lambda^*(\tau) = \infty$. However, because of saturation of $\tanh$ activations (i.e. $\sup_{x \in \mathbb{R}} |\tanh(x)| = 1$) in Equation 4

   $$\lim_{\tau \to \infty} \Lambda^*(\tau) = \lim_{\tau \to \infty} f(\tau) < \text{softplus}\left(\sum_{d=1}^D |w_d^{(3)}| + b^{(3)}\right) < \infty$$

   Therefore, the PDF doesn't integrate to 1.

2. The FullyNN model assigns a non-zero amount of probability mass to the $(-\infty, 0)$ interval, which violates the assumption that inter-event times are strictly positive.

   Since the inter-event times $\tau$ are assumed to be strictly positive almost surely, it must hold that $\text{Prob}(\tau \leq 0) = F^*(0) = 0$, or equivalently $\Lambda^*(0) = 0$. However, we can see that

   $$\Lambda^*(0) = f(0) = \text{softplus}(\boldsymbol{W}^{(3)} \tanh(\boldsymbol{W}^{(2)} \tanh(\tilde{\boldsymbol{b}}^{(1)}) + \boldsymbol{b}^{(2)}) + \boldsymbol{b}^{(3)}) > 0$$

   which means that the FullyNN model permits negative inter-event times.

# D    IMPLEMENTATION DETAILS

## D.1    SHARED ARCHITECTURE

We implement SOSFlow, DSFlow and LogNormMix, together with baselines: RMTPP (Gompertz distribution), exponential distribution and a FullyNN model. All of them share the same pipeline, from the data preprocessing to the parameter tuning and model selection, differing only in the way we calculate $p^*(\tau)$. This way we ensure a fair evaluation. Our implementation uses Pytorch.[3]

From arival times $t_i$ we calculate the inter-event times $\tau_i = t_i - t_{i-1}$. Since they can contain very large values, RNN takes log-transformed and centered inter-event time and produces $\boldsymbol{h}_i \in \mathbb{R}^H$. In case we have marks, we additionally input $m_i$ — the index of the mark class from which we get mark embedding vector $\boldsymbol{m}_i$. In some experiments we use extra conditional information, such as metadata $\boldsymbol{y}_i$ and sequence embedding $\boldsymbol{e}_j$, where $j$ is the index of the sequence.

As illustrated in Section 3.3 we generate the parameters $\boldsymbol{\theta}$ of the distribution $p^*(\tau_i)$ from $[\boldsymbol{h}_i||\boldsymbol{y}_i||\boldsymbol{e}_j]$ using an affine layer. We apply a transformation of the parameters to enforce the constraints, if necessary.

All decoders are implemented using a common framework relying on normalizing flows. By defining the base distribution $q(z)$ and the inverse transformation $(g_1^{-1} \circ \cdots \circ g_M^{-1})$ we can evaluate the PDF $p^*(\tau)$ at any $\tau$, which allows us to train with maximum likelihood (Section 3.1).

---

[3]`https://pytorch.org/` (Paszke et al., 2017)

## D.2 LOG-NORMAL MIXTURE

The log-normal mixture distribution is defined in Equation 2. We generate the parameters of the distribution $\boldsymbol{w} \in \mathbb{R}^K, \boldsymbol{\mu} \in \mathbb{R}^K, \boldsymbol{s} \in \mathbb{R}^K$ (subject to $\sum_k w_k = 1, w_k \geq 0$ and $s_k > 0$), using an affine transformation (Equation 3). The log-normal mixture is equivalent to the following normalizing flow model

$$z_1 \sim \mathrm{GaussianMixture}(\boldsymbol{w}, \boldsymbol{\mu}, \boldsymbol{s})$$
$$z_2 = az_1 + b$$
$$\tau = \exp(z_2)$$

By using the affine transformation $z_2 = az_1 + b$ before the $\exp$ transformation, we obtain a better initialization, and thus faster convergence. This is similar to the batch normalization flow layer (Dinh et al., 2017), except that $b = \frac{1}{N} \sum_{i=1}^{N} \log \tau_i$ and $a = \sqrt{\frac{1}{N} \sum_{i=1}^{N} (\log \tau_i - b)}$ are estimated using the entire dataset, not using batches.

*Forward* direction samples a value from a Gaussian mixture, applies an affine transformation and applies $\exp$. In the *bacward* direction we apply log-transformation to an observed data, center it with an affine layer and compute the density under the Gaussian mixture.

## D.3 BASELINES

We implement FullyNN model (Omi et al., 2019) as described in Appendix C, using the official implementation as a reference[4]. The model uses feed-forward neural network with non-negative weights (enforced by clipping values at $0$ after every gradient step). Output of the network is a cumulative intensity function $\Lambda^*(\tau)$ from which we can easily get intensity function $\lambda^*(\tau)$ as a derivative w.r.t. $\tau$ using automatic differentiation in Pytorch. We get the PDF as $p^*(\tau) = \lambda^*(\tau) \exp(-\Lambda^*(\tau))$.

We implement RMTPP / Gompertz distribution (Du et al., 2016)[5] and the exponential distribution (Upadhyay et al., 2018) models as described in Appendix B.

All of the above methods define the distribution $p^*(\tau)$. Since the inter-event times may come at very different scales, we apply a linear scaling $\tilde{\tau} = a\tau$, where $a = \frac{1}{N} \sum_{i=1}^{N} \tau_i$ is estimated from the data. This ensures a good initialization for all models and speeds up training.

## D.4 DEEP SIGMOIDAL FLOW

A single layer of DSFlow model is defined as

$$f_{\boldsymbol{\theta}}^{DSF}(x) = \sigma^{-1} \left( \sum_{k=1}^{K} w_k \sigma \left( \frac{x - \mu_k}{s_k} \right) \right)$$

with parameters $\boldsymbol{\theta} = \{\boldsymbol{w} \in \mathbb{R}^K, \boldsymbol{\mu} \in \mathbb{R}^K, \boldsymbol{s} \in \mathbb{R}^K\}$ (subject to $\sum_k w_k = 1, w_k \geq 0$ and $s_k > 0$). We obtain the parameters of each layer using Equation 3.

We define $p(\tau)$ through the inverse transformation $(g_1^{-1} \circ \cdots \circ g_M^{-1})$, as described in Section 3.1.

$$z_M = g_M^{-1}(\tau) = \log \tau$$
$$\cdots$$
$$z_m = g_m^{-1}(z_{m+1}) = f_{\boldsymbol{\theta}_m}^{DSF}(z_{m+1})$$
$$\cdots$$
$$z_1 = \sigma(z_2)$$
$$z_1 \sim q_1(z_1) = \mathrm{Uniform}(0, 1)$$

We use the the batch normalization flow layer (Dinh et al., 2017) between every pair of consecutive layers, which significantly speeds up convergence.

---

[4] https://github.com/omitakahiro/NeuralNetworkPointProcess
[5] https://github.com/musically-ut/tf_rmtpp

## D.5 Sum-of-squares polynomial flow

A single layer of SOSFlow model is defined as

$$f^{SOS}(x) = a_0 + \sum_{k=1}^{K} \sum_{p=0}^{R} \sum_{q=0}^{R} \frac{a_{p,k} a_{q,k}}{p+q+1} x^{p+q+1}$$

There are no constraints on the polynomial coefficients $\boldsymbol{a} \in \mathbb{R}^{(R+1) \times K}$. We obtain $\boldsymbol{a}$ similarly to Equation 3 as $\boldsymbol{a} = \boldsymbol{V_a c} + \boldsymbol{b_a}$, where $\boldsymbol{c}$ is the context vector.

We define $p(\tau)$ by through the inverse transformation $(g_1^{-1} \circ \cdots \circ g_M^{-1})$, as described in Section 3.1.

$$z_M = g_M^{-1}(\tau) = \log \tau$$
$$\cdots$$
$$z_m = g_m^{-1}(z_{m+1}) = f_{\boldsymbol{\theta}_m}^{SOS}(z_{m+1})$$
$$\cdots$$
$$z_1 = \sigma(z_2)$$
$$z_1 \sim q_1(z_1) = \text{Uniform}(0, 1)$$

Same as for DSFlow, we use the the batch normalization flow layer between every pair of consecutive layers. When implementing SOSFlow, we used Pyro[6] for reference.

## D.6 Reparametrization sampling

Using a log-normal mixture model allows us to sample with reparametrization which proves to be useful, e.g. when imputing missing data (Section 5.4). In a score function estimator (Williams, 1992) given a random variable $x \sim p_\theta(x)$, where $\theta$ are parameters, we can compute $\nabla_\theta \mathbb{E}_{x \sim p_\theta(x)}[f(x)]$ as $\mathbb{E}_{x \sim p_\theta(x)}[f(x) \nabla_\theta \log p_\theta(x)]$. This is an unbiased estimator of the gradients but it often suffers from high variance. If the function $f$ is differentiable, we can obtain an alternative estimator using the reparametrization trick: $\epsilon \sim q(\epsilon), x = g_\theta(\epsilon)$. Thanks to this reparametrization, we can compute $\nabla_\theta \mathbb{E}_{x \sim p_\theta(x)}[f(x)] = \mathbb{E}_{\epsilon \sim q(\epsilon)}[\nabla_\theta f(g_\theta(\epsilon))]$. Such reparametrization estimator typically has lower variance than the score function estimator (Mohamed et al., 2019). In both cases, we estimate the expectation using Monte Carlo.

To sample with reparametrization from the mixture model we use the Straight-Through Gumbel Estimator (Jang et al., 2017). We first obtain a relaxed sample $\boldsymbol{z}^* = \text{softmax}((\log \boldsymbol{w} + \boldsymbol{o})/T)$, where each $o_i$ is sampled i.i.d. from a Gumbel distribution with zero mean and unit scale, and $T$ is the temperature parameter. Finally, we get a one-hot sample $\boldsymbol{z} = \text{onehot}(\arg \max_k z_k^*)$. While a discrete $\boldsymbol{z}$ is used in the forward pass, during the backward pass the gradients will flow through the differentiable $\boldsymbol{z}^*$.

The gradients obtained by the Straight-Through Gumbel Estimator are slightly biased, which in practice doesn't have a significant effect on the model's performance. There exist alternatives (Tucker et al., 2017; Grathwohl et al., 2018) that provide unbiased gradients, but are more expensive to compute.

## E  Dataset statistics

### E.1  Synthetic data

Synthetic data is generated according to Omi et al. (2019) using well known point processes. We sample 64 sequences for each process, each sequence containing 1024 events.

**Poisson.** Conditional intensity function for a homogeneous (or stationary) Poisson point process is given as $\lambda^*(t) = 1$. Constant intensity corresponds to exponential distribution.

**Renewal.** A stationary process defined by a log-normal probability density function $p(\tau)$, where we set the parameters to be $\mu = 1.0$ and $\sigma = 6.0$. Sequences appear clustered.

---

[6]https://pyro.ai/ (Bingham et al., 2018)

| Dataset name | Number of sequences | Number of events |
|---|---|---|
| LastFM | 929 | 1268385 |
| Reddit | 10000 | 672350 |
| Stack Overflow | 6633 | 480414 |
| MOOC | 7047 | 396633 |
| Wikipedia | 1000 | 157471 |
| Yelp | 300 | 215146 |

Table 2: Dataset statistics.

**Self-correcting.** Unlike the previous two, this point process depends on the history and is defined by a conditional intensity function $\lambda^*(t) = \exp(t - \sum_{t_i < t} 1)$. After every new event the intensity suddenly drops, inhibiting the future points. The resulting point patterns appear regular.

**Hawkes.** We use a self-exciting point process with a conditional intensity function given as $\lambda^*(t) = \mu + \sum_{t_i < t} \sum_{j=1}^{M} \alpha_j \beta_j \exp(-\beta_j(t - t_i))$. As per Omi et al. (2019), we create two different datasets: **Hawkes1** with $M = 1$, $\mu = 0.02$, $\alpha_1 = 0.8$ and $\beta_1 = 1.0$; and **Hawkes2** with $M = 2$, $\mu = 0.2$, $\alpha_1 = 0.4$, $\beta_1 = 1.0$, $\alpha_2 = 0.4$ and $\beta_2 = 20$. For the imputation experiment we use Hawkes1 to generate the data and remove some of the events.

### E.2 REAL-WORLD DATA

In addition we use real-world datasets that are described bellow. Table 2 shows their summary. All datasets have a large amount of unique sequences and the number of events per sequence varies a lot. Using marked temporal point processes to predict the type of an event is feasible for some datasets (e.g. when the number of classes is low), and is meaningless for other.

**LastFM.**[7] The dataset contains sequences of songs that selected users listen over time. Artists are used as an event type.

**Reddit.**[8] On this social network website users submit posts to subreddits. In the dataset, most active subreddits are selected, and posts from the most active users on those subreddits are recodered. Each sequence corresponds to a list of submissions a user makes. The data contains 984 unique subreddits that we use as classes in mark prediction.

**Stack Overflow.**[9] Users of a question-answering website get rewards (called badges) over time for participation. A sequence contains a list of rewards for each user. Only the most active users are selected and only those badges that users can get more than once.

**MOOC.**[8] Contains the interaction of students with an online course system. An interaction is an event and can be of various types (97 unique types), e.g. watching a video, solving a quiz etc.

**Wikipedia.**[8] A sequence corresponds to edits of a Wikipedia page. The dataset contains most edited pages and users that have an activity (number of edits) above a certain threshold.

**Yelp.**[10] We use the data from the review forum and consider the reviews for the 300 most visited restaurants in Toronto. Each restaurant then has a corresponding sequence of reviews over time.

## F  ADDITIONAL DISCUSSION OF THE EXPERIMENTS

### F.1  EVENT TIME PREDICTION USING HISTORY

**Detailed setup.** Each dataset consists of multiple sequences of inter-event times. We consider 10 train/validation/test splits of the sequences (of sizes $60\%/20\%/20\%$). We train all model parameters by minimizing the negative log-likelihood (NLL) of the training sequences, defined as $\mathcal{L}_{time}(\boldsymbol{\theta}) =$

---

[7]Celma (2010)

[8]https://github.com/srijankr/jodie/(Kumar et al., 2019)

[9]https://archive.org/details/stackexchange preprocessed according to Du et al. (2016)

[10]https://www.yelp.com/dataset/challenge

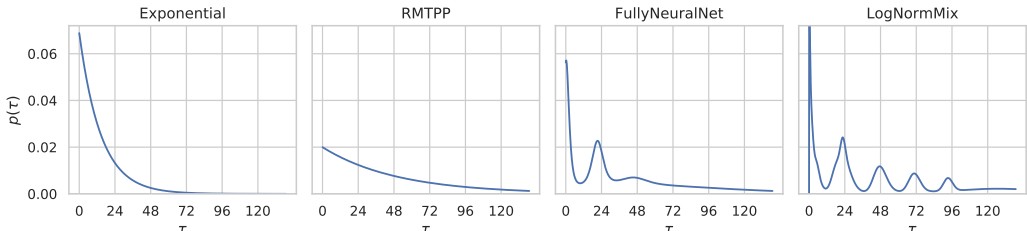

Figure 9: Models learn different conditional distribution $p(\tau|\mathcal{H})$ on Yelp dataset. Since check-ins occur during the opening hours, true distribution of the next check-in resembles the one on the right.

$-\frac{1}{N}\sum_{i=1}^{N}\log p_{\boldsymbol{\theta}}^*(\tau_i)$. After splitting the data into the 3 sets, we break down long training sequences into sequences of length at most 128. Optimization is performed using Adam (Kingma & Ba, 2015) with learning rate $10^{-3}$. We perform training using mini-batches of 64 sequences. We train for up to 2000 epochs (1 epoch = 1 full pass through all the training sequences). For all models, we compute the validation loss at every epoch. If there is no improvement for 100 epochs, we stop optimization and revert to the model parameters with the lowest validation loss.

We select hyperparameter configuration for each model that achieves the lowest average loss on the validation set. For each model, we consider different values of $L_2$ regularization strength $C \in \{0, 10^{-5}, 10^{-3}\}$. Additionally, for SOSFlow we tune the number of transformation layers $M \in \{1, 2, 3\}$ and for DSFlow $M \in \{1, 2, 3, 5, 10\}$. We have chosen the values of K such that the mixture model has approximately the same number of parameters as a 1-layer DSFlow or a 1-layer FullyNN model. More specifically, we set $K = 64$ for LogNormMix, DSFlow and FullyNN. We found all these models to be rather robust to the choice of $K$, as can be seen in Table 3 for LogNormMix. For SOSFlow we used $K = 4$ and $R = 3$, resulting in a polynomial of degree 7 (per each layer). Higher values of $R$ led to unstable training, even when using batch normalization.

**Additional discussion.** In this experiment, we only condition the distribution $p^*(\tau_i)$ on the history embedding $\boldsymbol{h}_i$. We don't learn sequence embeddings $\boldsymbol{e}_j$ since they can only be learned for the training sequences, and not fore the validation/test sets.

There are two important aspects related to the NLL loss values that we report. First, the absolute loss values can be arbitrarily shifted by rescaling the data. Assume, that we have a distribution $p(\tau)$ that models the distribution of $\tau$. Now assume that we are interested in the distribution $q(x)$ of $x = a\tau$ (for $a > 0$). Using the change of variables formula, we obtain $\log q(x) = \log p(\tau) + \log a$. This means that by simply scaling the data we can arbitrarily offset the log-likelihood score that we obtain. Therefore, the absolute values of of the (negative) log-likelihood $\mathcal{L}$ for different models are of little interest — all that matters are the differences between them.

The loss values are dependent on the train/val/test split. Assume that model 1 achieves loss values $\mathcal{L}_1 = \{1.0, 3.0\}$ on two train/val/test splits, and model 2 achieves $\mathcal{L}_2 = \{2.0, 4.0\}$ on the same splits. If we first aggregate the scores and report the average $\hat{\mathcal{L}}_1 = 2.0 \pm 1.0$, $\hat{\mathcal{L}}_2 = 3.0 \pm 1.0$, it may seem that the difference between the two models is not significant. However, if we first compute the differences and then aggregate $(\mathcal{L}_2 - \mathcal{L}_1) = 1.0 \pm 0.0$ we see a different picture. Therefore, we use the latter strategy in Figure 3. For completeness, we also report the numbers obtained using the first strategy in Table 4.

As a baseline, we also considered the constant intensity / exponential distribution model (Upadhyay et al., 2018). However, we excluded the results for it from Figure 3, since it consistently achieved the worst loss values and had high variance. We still include the results for the constant intensity model in Table 4. We also performed all the experiments on the synthetic datasets (Appendix E.1). The results are shown in Table 5, together with NLL scores under the true model. We see that LogNormMix and DSFlow, besides achieving the best results, recover the true distribution.

Finally, in Figure 9 we plot the conditional distribution $p(\tau|\mathcal{H})$ with models trained on Yelp dataset. The events represent check-ins into a specific restaurant. Since check-ins mostly happen during the opening hours, the inter-event time is likely to be on the same day (0h), next day (24h), the day after (48h), etc. LogNormMix can fully recover this behavior from data while others either cannot learn multimodal distributions (e.g. RMTPP) or struggle to capture it (e.g. FullyNN).

| $K$ | 2 | 4 | 8 | 16 | 32 | 64 |
|---|---|---|---|---|---|---|
| Reddit | 10.239 | 10.208 | 10.189 | 10.185 | 10.191 | 10.192 |
| LastFM | -2.828 | -2.879 | -2.881 | -2.880 | -2.877 | -2.860 |
| MOOC | 6.246 | 6.053 | 6.055 | 6.055 | 6.050 | 5.660 |
| Stack Overflow | 14.461 | 14.438 | 14.435 | 14.435 | 14.436 | 14.428 |
| Wikipedia | 8.399 | 8.389 | 8.385 | 8.384 | 8.384 | 8.386 |
| Yelp | 13.169 | 13.103 | 13.058 | 13.045 | 13.032 | 13.024 |
| Poisson | 1.006 | 0.992 | 0.991 | 0.991 | 0.990 | 0.991 |
| Renewal | 0.256 | 0.254 | 0.254 | 0.254 | 0.256 | 0.259 |
| Self-correcting | 0.831 | 0.785 | 0.782 | 0.783 | 0.784 | 0.784 |
| Hawkes1 | 0.530 | 0.523 | 0.532 | 0.532 | 0.523 | 0.523 |
| Hawkes2 | 0.036 | 0.026 | 0.024 | 0.024 | 0.026 | 0.024 |

Table 3: Performance of LogNormMix model for different numbers $K$ of mixture components.

| | Reddit | LastFM | MOOC | Stack Overflow | Wikipedia | Yelp |
|---|---|---|---|---|---|---|
| LogNormMix | **10.19 ± 0.078** | **-2.88 ± 0.147** | **6.03 ± 0.092** | **14.44 ± 0.013** | **8.39 ± 0.079** | **13.02 ± 0.070** |
| DSFlow | 10.20 ± 0.074 | **-2.88 ± 0.148** | **6.03 ± 0.090** | **14.44 ± 0.019** | 8.40 ± 0.090 | 13.09 ± 0.065 |
| SOSFlow | 10.27 ± 0.106 | -2.56 ± 0.133 | 6.27 ± 0.058 | 14.47 ± 0.049 | 8.44 ± 0.120 | 13.21 ± 0.068 |
| FullyNN | 10.23 ± 0.072 | -2.84 ± 0.179 | 6.83 ± 0.152 | 14.45 ± 0.014 | 8.40 ± 0.086 | 13.04 ± 0.073 |
| LogNormal | 10.38 ± 0.077 | -2.60 ± 0.140 | 6.53 ± 0.016 | 14.62 ± 0.013 | 8.52 ± 0.078 | 13.44 ± 0.074 |
| RMTPP | 10.88 ± 0.293 | -1.30 ± 0.164 | 10.65 ± 0.023 | 14.51 ± 0.014 | 10.02 ± 0.085 | 13.36 ± 0.056 |
| Exponential | 11.07 ± 0.070 | -1.28 ± 0.152 | 10.64 ± 0.026 | 18.48 ± 3.257 | 10.03 ± 0.083 | 13.78 ± 1.250 |

Table 4: Time prediction test NLL on real-world data.

## F.2 LEARNING WITH MARKS

**Detailed setup.** We use the same setup as in Section F.1, except two differences. For learning in a marked temporal point process, we mimic the architecture from Du et al. (2016). The RNN takes a tuple $(\tau_i, m_i)$ as input at each time step, where $m_i$ is the mark. Moreover, the loss function now includes a term for predicting the next mark: $\mathcal{L}_{total}(\boldsymbol{\theta}) = -\frac{1}{N} \sum_{i=1}^{N} [\log p_{\boldsymbol{\theta}}^*(\tau_i) + \log p_{\boldsymbol{\theta}}^*(m_i)]$.

The next mark $m_i$ at time $t_i$ is predicted using a categorical distribution $p^*(m_i)$. The distribution is parametrized by the vector $\boldsymbol{\pi}_i$, where $\pi_{i,c}$ is the probability of event $m_i = c$. We obtain $\boldsymbol{\pi}_i$ using the history embedding $\boldsymbol{h}_i$ passed through a feedforward neural network

$$\boldsymbol{\pi}_i = \text{softmax}\left(\boldsymbol{V}_{\boldsymbol{\pi}}^{(2)} \tanh(\boldsymbol{V}_{\boldsymbol{\pi}}^{(1)} \boldsymbol{h}_i + \boldsymbol{b}_{\boldsymbol{\pi}}^{(1)}) + \boldsymbol{b}_{\boldsymbol{\pi}}^{(2)}\right)$$

where $\boldsymbol{V}_{\boldsymbol{\pi}}^{(1)}, \boldsymbol{V}_{\boldsymbol{\pi}}^{(2)} \boldsymbol{b}_{\boldsymbol{\pi}}^{(1)}, \boldsymbol{b}_{\boldsymbol{\pi}}^{(2)}$ are the parameters of the neural network.

**Additional discussion.** In Figure 3 (right) we reported the differences in time NLL between different models $\mathcal{L}_{time}(\boldsymbol{\theta}) = -\frac{1}{N} \sum_{i=1}^{N} \log p_{\boldsymbol{\theta}}^*(\tau_i)$. In Table 6 we additionally provide the total NLL $\mathcal{L}_{total}(\boldsymbol{\theta}) = -\frac{1}{N} \sum_{i=1}^{N} [\log p_{\boldsymbol{\theta}}^*(\tau_i) + \log p_{\boldsymbol{\theta}}^*(m_i)]$ averaged over multiple splits.

Using marks as input to the RNN improves time prediction quality for all the models. However, since we assume that the marks are conditionally independent of the time given the history (as was done in earlier works), all models have similar mark prediction accuracy.

## F.3 LEARNING WITH ADDITIONAL CONDITIONAL INFORMATION

**Detailed setup.** In the Yelp dataset, the task is to predict the time $\tau_i$ until the next customer check-in, given the history of check-ins up until the current time $t_{i-1}$. We want to verify our intuition that the distribution $p^*(\tau_i)$ depends on the current time $t_{i-1}$. For example, $p^*(\tau_i)$ might be different depending on whether it's a weekday and / or it's an evening hour. Unfortunately, a model that processes the history with an RNN cannot easily obtain this information. Therefore, we provide this information directly as a context vector $\boldsymbol{y}_i$ when modeling $p^*(\tau_i)$.

The first entry of context vector $\boldsymbol{y}_i \in \{0, 1\}^2$ indicates whether the previous event $t_{i-1}$ took place on a weekday or a weekend, and the second entry indicates whether $t_{i-1}$ was in the 5PM–11PM time

| | Poisson | Renewal | Self-correcting | Hawkes1 | Hawkes2 |
|---|---|---|---|---|---|
| True model | 0.999 | 0.254 | 0.757 | 0.453 | -0.043 |
| LogNormMix | **0.99 ± 0.006** | **0.25 ± 0.010** | **0.78 ± 0.003** | **0.52 ± 0.047** | **0.02 ± 0.049** |
| DSFlow | **0.99 ± 0.006** | **0.25 ± 0.010** | **0.78 ± 0.002** | **0.52 ± 0.047** | **0.02 ± 0.050** |
| SOSFlow | 1.00 ± 0.013 | **0.25 ± 0.010** | 0.88 ± 0.011 | 0.59 ± 0.056 | 0.06 ± 0.046 |
| FullyNN | 1.00 ± 0.006 | 0.28 ± 0.013 | **0.78 ± 0.004** | 0.55 ± 0.047 | 0.06 ± 0.047 |
| LogNormal | 1.08 ± 0.008 | **0.25 ± 0.010** | 1.03 ± 0.006 | 0.55 ± 0.047 | 0.06 ± 0.049 |
| RMTPP | **0.99 ± 0.006** | 1.01 ± 0.023 | **0.78 ± 0.003** | 0.74 ± 0.057 | 0.69 ± 0.058 |
| Exponential | **0.99 ± 0.006** | 1.00 ± 0.023 | 0.94 ± 0.002 | 0.74 ± 0.055 | 0.69 ± 0.054 |

Table 5: Time prediction test NLL on synthetic data.

| | Time NLL | | Total NLL | | Mark accuracy | |
|---|---|---|---|---|---|---|
| | Reddit | MOOC | Reddit | MOOC | Reddit | MOOC |
| LogNormMix | **10.28 ± 0.066** | **5.75 ± 0.040** | 12.40 ± 0.094 | 7.58 ± 0.047 | 0.62±0.014 | 0.45±0.003 |
| DSFlow | **10.28 ± 0.073** | 5.78 ± 0.067 | **12.39 ± 0.064** | **7.52 ± 0.074** | 0.62±0.013 | 0.45±0.004 |
| SOSFlow | 10.35 ± 0.106 | 6.06 ± 0.084 | 12.49 ± 0.158 | 7.78 ± 0.107 | 0.62±0.013 | **0.46±0.009** |
| FullyNN | 10.41 ± 0.079 | 6.22 ± 0.224 | 12.51 ± 0.094 | 7.93 ± 0.230 | **0.63±0.013** | **0.46±0.004** |
| LogNormal | 10.42 ± 0.076 | 6.38 ± 0.019 | 12.51 ± 0.080 | 8.11 ± 0.026 | 0.62±0.013 | 0.42±0.005 |
| RMTPP | 11.15 ± 0.061 | 10.29 ± 0.209 | 13.26 ± 0.085 | 12.14 ± 0.220 | 0.62±0.014 | 0.41±0.006 |

Table 6: Time and total NLL and mark accuracy when learning a marked TPP.

window. To each of the four possibilities we assign a learnable 64-dimensional embedding vector. The distribution of $p^*(\tau_i)$ until the next event depends on the embedding vector of the time stamp $t_{i-1}$ of the most recent event.

## F.4 MISSING DATA IMPUTATION

**Detailed setup.** The dataset for the experiment is generated as a two step process: 1) We generate a sequence of 100 events from the model used for Hawkes1 dataset (Appendix E.1) resulting in a sequence of arrival times $\{t_1, \ldots t_N\}$, 2) We choose random $t_i$ and remove all the events that fall inside the interval $[t_i, t_{i+k}]$ where $k$ is selected such that the interval length is approximately $t_N/3$.

We consider three strategies for learning with missing data (shown in Figure 4 (left)):

a) **No imputation**. The missing block spans the time interval $[t_i, t_{i+k}]$. We simply ignore the missing data, i.e. training objective $\mathcal{L}_{time}$ will include an inter-event time $\tau = t_{i+k} - t_i$.

b) **Mean imputation**. We estimate the average inter-event time $\hat{\tau}$ from the observed data, and impute events at times $\{t_i + n\hat{\tau} \;$ for $\; n \in \mathbb{N}$, such that $\; t_i + n\hat{\tau} < t_{i+k}\}$. These imputed events are fed into the history-encoding RNN, but are not part of the training objective.

c) **Sampling** . The RNN encodes the history up to and including $t_i$ and produces $\boldsymbol{h}_i$ that we use to define the distribution $p^*(\tau|\boldsymbol{h}_i)$. We draw a sample $\tau_j^{(imp)}$ form this distribution and feed it into the RNN. We keep repeating this procedure until the samples get past the point $t_{i+k}$. The imputed inter-event times $\tau_j^{(imp)}$ are affecting the hidden state of the RNN (thus influencing the likelihood of future observed inter-event times $\tau_i^{(obs)}$).

We sample multiple such sequences in order to approximate the *expected* log-likelihood of the observed inter-event times $\mathbb{E}_{\tau^{(imp)} \sim p^*} \left[ \sum_i \log p^*(\tau_i^{(obs)}) \right]$. Since this objective includes an expectation that depends on $p^*$, we make use of reparametrization sampling to obtain the gradients w.r.t. the distribution parameters (Mohamed et al., 2019).

## F.5 SEQUENCE EMBEDDING

**Detailed setup.** When learning sequence embeddings, we train the model as described in Appendix F.1, besides one difference. First, we pre-train the sequence embeddings $\boldsymbol{e}_j$ by disabling the his-

tory embedding $\boldsymbol{h}_i$ and optimizing $-\frac{1}{N}\sum_i \log p_{\boldsymbol{\theta}}(\tau_i|\boldsymbol{e}_j)$. Afterwards, we enable the history and minimize $-\frac{1}{N}\sum_i \log p_{\boldsymbol{\theta}}(\tau_i|\boldsymbol{e}_j, \boldsymbol{h}_i)$.

In Figure 6 the top row shows samples generated using $\boldsymbol{e}_{SC}$, embedding of a self-correcting sequence, the bottom row was generated using $\boldsymbol{e}_{SC}$, embedding of a renewal sequence, and the middle row was generated using $1/2(\boldsymbol{e}_{SC} + \boldsymbol{e}_{RN})$, an average of the two embeddings.

