# OpenReview forum: "Intensity-Free Learning of Temporal Point Processes"
_ICLR.cc/2020/Conference — Accept (Spotlight)_

### Official Review · AnonReviewer2 · 2019-10-22
**Official Blind Review #2**

**Rating:** 8

**Review:**

The authors propose a new paradigm for learning models for point processes which circumvents the need to explicitly model the conditional intensity. This utilizes recent work on Normalizing flows and upends a long-standing paradigm. The paper is a true tour-de-force and the authors make a very convincing case for why instead of modelling conditional intensity, one could (and should) model the distribution of times directly.

The authors do an extensive literature review and place their work in the context very well. The writing is very lucid and the paper was a pleasure to read. The illustrations are also helpful and the extensive experiments complement the discussions very well. The appendix is also very easy to read and has been judiciously separated.

The two primary axes they justify their model is by first comparing it against conditional intensity-based methods in general and show that:
 1. Their approach has universal approximation property (thanks to a result from DasGupta (2008)).
 2. The likelihood function can be evaluated efficiently (thanks to it being a mixture of log-normals).
 3. The expected next action time can be evaluated efficiently.
 4. Sampling from their model is also easy.

Then they show that all related work fails either one or more of these conditions. Finally, they also address the common concerns of why point processes were historically always taught using intensity functions. A bit of historical context here is that in absence of modern computers and when Cox-Hazard models still needed to be worked out by hand, the conditional intensity functions indeed were much simpler to handle and afforded desirable properties. However, with modern approaches towards modelling probability distributions, those reasons are no longer valid. Further, some of the recent research which has built on the conditional intensity functions has lost much of the advantages anyway because of the use of deep neural networks. Hence, at this point, one might as well move directly to modelling the probability (as the authors have done) instead of taking a detour via the intensity functions. The experimental results for prediction seem to justify taking this route as modelling intensity function does not seem to out-perform the intensity-free models (more on the metrics later). Finally, the authors also show that their model, thanks to its generative nature, can also be used for several ancillary tasks: Sequence embedding, missing data imputation and learning with conditional information. The provided code is easy to read through and execute.

Hence, I do not have any hesitation in recommending the paper for publication.

As a side-note, it is notable that in (Tabibian, 2016), a mixture model is proposed, but I believe it is not for the final probability but instead for the intensity itself to make the learning problem tractable.


Some ways of improving the paper:

 - Theorem 1 can be made more rigorous by making the role of parameters like 'K' more explicit. e.g. does the theorem imply that there 'exist' K, \mu_k, \s_k, or does it say that for any K, \mu_k, and \s_k, such 'w_k' can be found, etc.?
 - Using NLL for comparing models in the experiment results is a bit unusual. Could a different metric (e.g. MAE) be employed?
 - A key discussion missing in the paper is that of the complexity of training the model. RMTPP, for example, can be trained very efficiently while Neural-Hawkes is difficult to train due to a Monte-Carlo sampling embedded in the calculation of the likelihood. Empirical results will also add to better placing the Intensity free method against other methods.

Citations:

 Tabibian, Behzad, et al. "Distilling information reliability and source trustworthiness from digital traces." Proceedings of the 26th International Conference on World Wide Web. International World Wide Web Conferences Steering Committee, 2017.

**Experience Assessment:**

I have published in this field for several years.

**Review Assessment: Checking Correctness Of Derivations And Theory:**

I carefully checked the derivations and theory.

**Review Assessment: Checking Correctness Of Experiments:**

I carefully checked the experiments.

**Review Assessment: Thoroughness In Paper Reading:**

I read the paper thoroughly.

---

> ### Author Response · Authors · 2019-11-08
> **Response to Review #2**
>
> Thank you for your positive review and constructive comments.
>
>
> 1. We have made the statement of Theorem 1 more precise.
>
>
> 2. First, using MSE / MAE as a metric may discard important information about the learned distribution. For a model to achieve the best possible MSE, it is sufficient for the learned conditional distribution p_model(tau | history) to have the same mean as the true distribution p_data(tau | history). Similarly, MAE is minimized by matching the median. Unlike NLL (i.e. KL divergence), MSE & MAE do not reflect how well p_model captures other important properties of p_data, such as multimodality or tail behavior.
>
> Second, the expected inter-event time cannot be computed in closed form for some of the models that we considered (namely FullyNN, DSFlow and SOSFlow). Approximating the expectation via Monte Carlo is also non-trivial for these models, since they do not permit closed-form sampling. This makes computing MSE / MAE more challenging for these models.
>
> Last, the NLL scores make it easy to compare our models to the _true_ generative models for the synthetic datasets (Table 5). Since DSFlow and LogNormMix achieve nearly the same NLL loss on the test set as the true generative model, we can conclude that DSFlow and LogNormMix have successfully recovered the true data distribution.
>
>
> 3. We have measured the time it takes to perform a single forward pass for different models on the MOOC dataset (averaged over 200 trials).
>
> Model 		 | RNN time (ms) | Decoder time (ms) |
> RMTPP 	         | 3.91 | 0.66 |
> LogNormMix | 3.70 | 0.96 |
> FullyNN 	         | 3.68 | 2.06 |
> Exponential 	 | 3.68 | 0.47 |
> SOSFlow 	 | 3.79 | 3.18 |
> DSFlow 	         | 3.56 | 3.32 |
>
> We make 2 main observations. First, the RNN takes a significant portion of the runtime. Using a different architecture (e.g. LSTM) would increase this even further. Second, the overhead for using LogNormMix (with 64 mixture components) over RMTPP is minor, while the performance gains are significant.
>
>
> Also, we have added a reference to (Tabibian et al., 2016) to our paper.

---

### Official Review · AnonReviewer3 · 2019-10-23
**Official Blind Review #3**

**Rating:** 6

**Review:**

This paper describes a simple yet effective technique for learning temporal point processes using a mixture of log-normal densities whose parameters are estimated with neural networks that also adds conditional information. The method is shown to perform better than more recent techniques for density estimation such as different versions of normalising flows. Experiments were reported on 6 datasets, comparing the approach against flow models and assessing the benefits of adding extra conditional information, performance with missing data, and benefits of sequence embeddings.

The paper makes an important point which is also my own experience when working with relatively low dimensional problems; simpler neural density estimation approaches such as MDNs usually perform similarly or even better than models using normalising flows. The task here is on learning temporal point processes which have important applications in social networks, criminality studies, disease modelling, etc, but are relatively unpopular within the machine learning community. The paper gives some motivation but I think the authors could elaborate further on the huge number of applications and potential for significant impact from these models. Apart from this the paper is well written and structured, and easy to follow.

There are not many theoretical innovations as the main contribution is a combination of several well known techniques such as MDNs and RNNs, applied to the specific temporal point process formulation. The main lesson learnt though is that these simpler techniques can perform surprisingly well. With that said, the paper would benefit from discussing the following points:

1) Mixture models are effective in low dimensional problems but require the manual specification of the number of components. How was this done in the experiments and how sensitive the performance is to this parameter?

2) The paper discusses several problems with normalising flows, but in particular the computational cost involved in generating samples or evaluating the density. This is true for some variations of NFs but not for all. For example RealNVP and the recent Neural Spline Flows are efficient in both, sample generation and density evaluation. With this in mind, the paper would benefit from further comparisons to these approaches. Another interesting comparison would be with autoregressive flows, such as inverse autoregressive flow or masked autoregressive flow. They can both  capture sequences and should be able to model inter-event times, instead of an RNN.



**Experience Assessment:**

I have read many papers in this area.

**Review Assessment: Checking Correctness Of Derivations And Theory:**

I assessed the sensibility of the derivations and theory.

**Review Assessment: Checking Correctness Of Experiments:**

I assessed the sensibility of the experiments.

**Review Assessment: Thoroughness In Paper Reading:**

I read the paper at least twice and used my best judgement in assessing the paper.

---

> ### Author Response · Authors · 2019-11-08
> **Response to Review #3**
>
> Thank you for your positive review and constructive comments.
>
>
> 1) Overall, we found the results to be very robust to the choice of K. We have included the discussion in Appendix F.1 and an additional Table 3 showing the stability of the results for the mixture model when varying K.
>
> In the main experiments, we used K = 64 in order to make the number of parameters comparable between the mixture and single-layer DSFlow and FullyNN models. When manually inspecting some of the learned conditional distributions, we found that in this case most of the mixture components were assigned a weight w_k close to zero. This means that it's safe to use a large K, and the model will automatically only use the necessary number of components.
>
>
> 2a) Please note that RealNVP and Neural Spline Flow, while successfully used in multivariate density estimation, cannot be applied to 1D data, such as inter-event times.
>
> RealNVP and Neural Spline Flow define an invertible transformation by splitting the dimensions of the variable into two groups, where the first group remains unchanged and the second group is passed through an invertible transformation. In 1D, we cannot split the dimensions. Therefore, these architectures cannot be used in our scenario. We have clarified this point in the revised version of the paper.
>
>
> 2b) All the models considered in our work are autoregressive; SOSFlow model is a strict generalization of MAF / IAF.
>
> The MAF / IAF models specify the joint distribution $p(\tau_1, …, \tau_N)$ by employing an autoregressive factorization $p(\tau_1, …, \tau_N) = \prod_{i=1}^N p(\tau_i | \tau_1, …, \tau_{i-1})$.
>
> The two main components of such a model are
>  - An autoregressive neural network that takes $\{\tau_1, …, \tau_{i-1}\}$ as input and produces the transformation parameters $\theta_i$.
>  - A parametric transformation $f( . ; \theta)$  that transforms $\tau_i = f(z_i; \theta_i)$ for $z_i \sim N(0, 1)$, and thus defines the conditional distribution $p(\tau_i | \tau_1, …, \tau_{i-1})$.
>
> In the original IAF paper, the authors use an LSTM as the autoregressive NN that produces parameters of an affine transformation $f(z; a, b) = az + b$.
>
> The SOSFlow transformation used in our paper is a strict generalization of the affine transformation (as pointed out by the authors of the original paper [1]), therefore our model can already be seen as a generalization of the MAF / IAF architecture. We use RNN as an autoregressive NN instead of a more sophisticated architecture (e.g., LSTM, Wavenet, Transformer) since (1) this is the de-facto standard used in other neural temporal point process works [2, 3] and (2) to highlight that the improved performance of our model comes from a better model for the distribution of inter-event times, not because we are using a different conditioner network.
>
>
> [1] Jaini, P., et al., "Sum-of-Squares Polynomial Flow", ICML 2019
> [2] Du, N., et al., "Recurrent marked temporal point processes: Embedding event history to vector", KDD 2016
> [3] Omi, T., et al.. "Fully Neural Network based Model for General Temporal Point Processes", NeurIPS 2019

---

### Official Review · AnonReviewer1 · 2019-10-26
**Official Blind Review #1**

**Rating:** 8

**Review:**

The paper proposes to directly model the (conditional) inter-event intervals in a temporal point process, and demonstrates two different ways of parametrizing this distribution, one via normalizing flows and another via a log-normal mixture model. To increase the expressiveness of the resulting model, the parameters of these conditional distributions are made to be dependent on histories and additional input features through a RNN network, updating at discrete event occurrences.

The paper is very well written and easy to follow. I also like the fact that it is probably among the first of those trying to integrate neural networks into TPPs to look at directly modeling inter-event intervals, which offers a different perspective and potentially also opens doors for many new methods to come.

I have just three comments/questions.

1. The log-normal mixture model has a hyper-parameter K. Similarly, DSFlow also has K, and SOSFlow has K and R. How are these hyper-parameters selected? I don't seem to find any explanation in the paper (not even in appendix F.1)?

2. To better demonstrate that a more complicated (e.g. multi-modal) inter-event interval distribution is necessary and can really help with data modeling, I'd be interested to see e.g. those different interval distributions (learnt from different models) being plotted against each other (sth. similar to Figure 8, but with actual learnt distributions), and preferably with some meaningful explanations as to e.g. how the additional modes capture or reflect what we know about the data.

3. Even though the current paper mainly focuses on inter-event interval prediction, I think it's still helpful to also report the  model's prediction accuracy on marks in a MTPP. The "Total NLL" in Table 5 is one step towards that, but a separate performance metric on mark prediction alone would have been even clearer.

**Experience Assessment:**

I have published one or two papers in this area.

**Review Assessment: Checking Correctness Of Derivations And Theory:**

I assessed the sensibility of the derivations and theory.

**Review Assessment: Checking Correctness Of Experiments:**

I carefully checked the experiments.

**Review Assessment: Thoroughness In Paper Reading:**

I read the paper thoroughly.

---

> ### Author Response · Authors · 2019-11-08
> **Response to Review #1**
>
> Thank you for your positive review and constructive comments.
>
> 1. We have chosen the values of K such that the mixture model has approximately the same number of parameters as a 1-layer DSFlow or a 1-layer FullyNN model (models with M>1 layers have approximately M times more parameters, respectively). More specifically, we set K=64 for LogNormMix, DSFlow and FullyNN. We found all these models to be rather robust to the choice of K, and values in range [8, 64] produced similar results. For SOSFlow we used K = 4 and limited R to 3 (the resulting polynomial was of degree 7 x #layers), since higher values led to unstable training, even when using Batch Normalization.
>
> We have included these details to Appendix F.1.
>
>
> 2. Based on your suggestion, we have added a visualization and a discussion of the learned conditional density p(tau | history) in Figure 9, Appendix F.1.
>
> As an example, we consider the Yelp dataset. The events represent check-ins into a specific restaurant. Since check-ins mostly happen during the opening hours, the inter-event time is likely to be ~0h (on the same day), ~24h (next day), etc. Figure 9 shows that LogNormMix can fully recover this behavior from data while others either cannot learn multimodal distributions (e.g. RMTPP) or struggle to capture it (e.g. FullyNN).
>
> Having a flexible distribution is beneficial even if the true distribution is not multimodal. This statement is supported by our experiments on synthetic data (Table 5): flexible mixture and flow-based models can almost perfectly approximate different generative processes.
>
>
> 3. We have included a Table 6 that compares the mark prediction accuracy in Appendix F.2. The goal of the marked TPP experiment is to assess whether time prediction quality can be improved with the knowledge of marks. We predict marks in the same way as done by RMTPP, i.e. output distributions of mark types and event time are conditionally independent. Because of this, mark prediction performance is similar across all the models.

---

> > ### Comment · AnonReviewer1 · 2019-11-14
> > **A potential reason for why mark prediction performance is poorer than Neural Hawkes?**
> >
> > Thank you for the meticulous response and update.
> >
> > From the new "mark accuracy" column in Table 6, it seems as though all the models considered in this paper are relatively poor at jointly modeling marks and times in a marked TPP, compared to say Neural Hawkes Process, which models an intensity function for each type (I tend to think this unique design helps with mark prediction, more below) and is missing from the comparison.
> >
> > I understand your argument that the paper is primarily focused on time prediction and therefore marks are only used as additional inputs to help with that in the experiments. But it just came to me that maybe mark prediction in marked TPPs isn't (and probably shouldn't be) as simple as just stacking a softmax module for marks on top of a normal TPP and then "jointly" optimizing the two conditionally independent likelihoods. The reason is that the likelihood on "time" is continuous while that on "mark" is discrete, and therefore when you take log-likelihood and use simple summation to combine them in the objective, one ("time") can easily "outweigh" the other ("mark") simply because it enjoys a much higher "range". This is also clearly exemplified from the small difference between "Time NLL" and "Total NLL" in Table 6, which signals that "Mark NLL" is much smaller than "Time NLL" and is therefore causing mark prediction to be under-trained. This may be a universal problem for all those marked TPP models where marks and times are assumed conditionally independent given the history.

---

> > > ### Author Response · Authors · 2019-11-15
> > > **Mark prediction**
> > >
> > > As you correctly pointed out, we focused on event time prediction in this paper. We agree with you that mark prediction is an interesting research problem in its own right, and that the conditional independence assumption might be too restrictive in some cases.
> > >
> > > In theory, our model could be altered in several ways to improve marks prediction. First, our model is compatible with the “Neural Hawkes” approach --- we could directly model the distribution of K inter-event times (for each mark) separately and use a shared RNN to process the history. However, it’s unclear whether this will boost the performance (Figure 8 of the original Neural Hawkes paper reports performance comparable to RMTPP in event time and marks prediction).
> > >
> > > It could also be possible to improve the mark prediction performance of our model by removing the conditional independence assumption (e.g., by again using a mixture model: $p(\tau, m | \mathcal{H}) = \sum_z p(\tau | z, \mathcal{H}) p(m | z, \mathcal{H}) p(z)$).
> > >
> > > Regarding loss balancing: Time NLL can be shifted arbitrarily by scaling all the inter-event times by a positive factor (as can be seen from the change of variables formula), so it’s unclear whether time NLL indeed dominates marks NLL during training --- this issue deserves further investigation.

---

### Public Comment · ~Pranav_Poduval1 · 2020-05-13
**"We connect the fields of temporal point processes and neural density estimation", TPPs and Density estimation were already linked in prior works that you have not cited**

Shuai Xiao et al. work "Modeling The Intensity Function Of Point Process Via Recurrent Neural Networks" AAAI 2017 ( https://arxiv.org/abs/1705.08982 ) already modelled the conditional distribution of inter-event times as Gaussians, but it failed to handle the issue of negative times. "A Variational Auto-Encoder Model for Stochastic Point Processes" CVPR 2017 ( https://arxiv.org/abs/1904.03273 ) fixed this issue by fitting an exponential distribution with zero probability for negative times.  So your novelty should be extending it multimodal distribution using Normalizing Flows and the mixture of log-Normals.

Density estimation by definition can be unimodal or multimodal, so your claim of novelty "connecting the fields of temporal point processes and neural density estimation" is False. Basically, these two works have also managed to do intensity-Free Learning of Temporal Point Processes and rightfully deserve the credit for it.

Do note a contemporary paper of yours "Point Process Flows" ( https://openreview.net/forum?id=rklJ2CEYPH ) which is extremely similar to your work was rejected from ICLR 2020 for the same reason.

I hope you do add the required citations and clarify your novelty. Unless you fix this, it looks like you are hogging all the credit which is not rightfully due to you.

I am sorry if I seem rude, but I believe it is of great importance to give credit where it is due especially in academia. I understand if you do not want to continue this discussion, but I hope others who read this do not forget to cite the original work in upcoming works.

Also, do note I am not saying your paper should be rejected or anything like that. I simply hope you reshape your contribution because the current tone of the paper has managed to convince the reviewers that your work is causing a "paradigm shift" in the way TPPs are modelled which is simply not true.

I have deleted my comments from the thread and re-structured my criticism into a single flow to make it easier for everyone to understand my view.
Thank You.

---

> ### Author Response · Authors · 2020-05-14
> **Response from the authors**
>
> We disagree with your comment:
> 1. Every single TPP paper either implicitly or explicitly defines the conditional density $p(\tau | \mathcal{H})$. This is also true for older papers like Du et al. 2016 (to which we give credit for this idea). However, none of the papers explain how to make this distribution flexible AND tractable AND easy to sample from.
> 2. Modeling the conditional distribution of inter-event times with a Gaussian doesn't define a valid TPP. If we use such model for generation, we may get negative inter-event times, which doesn't make sense.

---

> ### Author Response · Authors · 2020-05-17
> **Updated authors' response to the edited comment**
>
> First of all, we find it extremely inappropriate that you attack us in the comments with completely unjustified claims, some of which you later deleted (including accusations like "blatant plagiarism"). Also, by constantly editing & deleting your comments and moving the goalposts you make it harder for other readers to judge the validity of your claims (in particular, you used our replies to adapt one of your incorrect initial statements, thus giving a wrong impression).
>
>
> Now, regarding the technical claims.
>
> Some background:
> - A temporal point process can be equivalently represented in terms of the conditional intensity function $\lambda^*(t)$, the conditional hazard function $h^*(\tau)$ or the conditional density function $p^*(\tau)$. Specifying any of them uniquely defines the other two. This is a classic result known at least since 1972 (Lewis, "Stochastic Point Processes", 1972) and used by virtually every paper that does maximum likelihood learning in TPPs. Obviously, we do not claim credit for this result, and neither is this result due to the papers that you mentioned.
>
> - The paper "Recurrent Marked Temporal Point Processes" (RMTPP) by Du et al. (2016) uses an RNN to embed the event history into a vector, and then uses the exponential intensity function, which corresponds to the Gompertz distribution (Equation 12 of the original paper). We clearly give credit to this seminal paper.
>
> As for the papers that you mentioned:
> - "Modeling The Intensity Function Of Point Process Via Recurrent Neural Networks" by Xiao et al. (2017) [1] replaces the Gompertz distribution with a Gaussian distribution with fixed variance. This does not only decrease the flexibility of the distribution, but also introduces the problem of allowing negative inter-event times, which leads to an ill-defined TPP.
>
> - "A Variational Auto-Encoder Model for Stochastic Point Processes" by Mehrasa et al. (2017) [2] models $p^*(\tau)$ with an exponential distribution, which corresponds to constant intensity. It is a well-known fact that exponential distribution is a special case of Gompertz distribution where the w parameter is set to 0. Since this is special case of the model by Du et al., we don't see why it deserves a special mention.
>
> Our work & claimed contributions:
> - Previous works that tried to increase the flexibility of TPPs have focused on using more flexible intensity functions. This lead either to intractable likelihoods (Mei & Eisner, 2017) and / or models where sampling & computing moments cannot be done in closed form (Omi et al. 2019). In our work, we point out that it's easier to increase the flexibility of the models by directly considering $p^*(\tau)$, since this allows us to re-use tools originally proposed in the context of neural density estimation. This way, we don't have to sacrifice tractability and ease of us, and are thus able to overcome the limitations of the existing approaches.
>
> This observation (that it's easier to define flexible and tractable TPPs via modeling $p^*(\tau)$ directly) was not done by the works [1] and [2] that you mentioned. In fact, [1] and [2] used conditional distributions that were even *less* flexible than in the original work by Du et al. We are not sure how these two works contribute to the problem that we are studying - how to define flexible & tractable TPPs?
>
> - We haven't used the phrase "paradigm shift" or anything similar in our paper.
>
> - The field of neural density estimation (https://arxiv.org/abs/1910.13233) is concerned with learning flexible probability distributions using neural networks. No papers that were publicly available at the submission time have discussed the connection of TPPs to mixture density networks or normalizing flows - two most prominent methods for neural density estimation. Therefore, we don't see a problem with the statement "We connect the fields of temporal point processes and neural density estimation".
>
> P.S.: Here is the screenshot of the post that we were replying to https://i.imgur.com/ftIHiYm.png (in case the posts gets edited / deleted yet again).

---

> > ### Public Comment · ~Pranav_Poduval1 · 2020-05-17
> > **I really don't want to continue this discussion, let others be the judge**
> >
> > The textbook definition of neural density estimation is "Neural density estimation is a parametric method for density estimation that uses neural networks to parameterize a density model". The density model can be as simple as Gaussians or exponential, so long we are using neural networks to find the optimal parameters of the distributions, by definition it falls under neural density estimation (Normalising flows and MoGs are fancy extensions to this). So yes I would consider claims like "We connect the fields of temporary point processes and neural density estimation", plagiarism. Since you are calling prior knowledge your own discovery. (Plagiarism definition - "the practice of taking someone else's work or ideas and passing them off as one's own").
> >
> > "We haven't used the phrase "paradigm shift" or anything similar in our paper"
> > Of course, I know you haven't, but due to the claims made in the paper ("We connect the fields of temporal point processes and neural density estimation") you have managed to convince the reviewers that your work is causing a "paradigm shift". Look at this way, the paper "Point Process Flow" ( https://openreview.net/pdf?id=rklJ2CEYPH ) never made such a wild claim and was rejected from the exact same conference ICLR 2020. because their work viewed as "incremental", though in essence, it's almost exactly same as your work, whereas you were given a spotlight, does that really make sense to you.
> >
> > "First of all, we find it extremely inappropriate that you attack us in the comments with completely unjustified claims, some of which you later deleted (including accusations like "blatant plagiarism"). Also, by constantly editing & deleting your comments and moving the goalposts you make it harder for other readers to judge the validity of your claims (in particular, you used our replies to adapt one of your incorrect initial statements, thus giving a wrong impression)."
> > I agree and apologise, the previous discussion thread was meant to be a discussion between us, since you said I was "rude" and that you were no longer going to reply. I saw the previous thread as pointless noise. So I re-structured all my criticism into a single flow to make it easy for future readers. Let them be the judge.
> >
> > Again I am sorry for "attacking" you. That wasn't my intention.
> >
> > Our views are biased we need external opinion for differentiating right from wrong. I have a strong feeling you have read the previous works and not citied them on purpose (especially the first one), or if you were unaware of the previous work, you still need to cite them and give them due credit once pointed out. But you believe yourself to be the first one to link TPPs and neural density estimation, eliminating the need for modelling intensity function.
> >
> > Our differences seem irreconcilable, so let's not make this another thread. Let others judge on our behalf.
> > Best of luck.

---

### Decision · Program_Chairs · 2019-12-19

**Decision:**

Accept (Spotlight)

**Comment:**

This submission proposes a new paradigm for modelling temporal point processes by using deep learning to learn to mix log-normal distributions in order to directly model the conditional distribution of event time intervals themselves.

Strengths of the paper:
-Introduces a new modelling paradigm that can lead to further research in this direction, for an important problem.
-Extensive experimentation validates the approach quantitatively.
-Easy to read.

Weaknesses:
-Several reviewers wanted more details on how the mixing parameter K was tuned. This was adequately addressed during the discussion period.

The reviewer consensus was to accept this submission.

---

> ### Public Comment · ~Pranav_Poduval1 · 2020-05-13
> **"Introduces a new modelling paradigm that can lead to further research in this direction, for an important problem" is False**
>
> I am not debating the acceptance of the paper, but I believe it is of gr8 importance to give credit where it is due especially in academia. It was Shuai Xiao et al. work "Modeling The Intensity Function Of Point Process Via Recurrent Neural Networks" (AAAI 2017) who first modelled conditional distribution of inter-event times as Gaussians, so the primary novelty of this work is extending it to a mixture of log-Normals and Normalizing Flows. It is possible the authors and the reviewers were unaware of the original work, but I believe it is our duty to ensure the original work is also cited along with this in future papers.
>
> Also a similar paper  "Point Process Flows" ICLR 2020 was rejected cuz of the same reason